# Preconditioning Neural Tangent Kernel for Adaptive Optimization

Xiyuan Yang [1]   Wenxuan Bao [1]   Katherine Tieu [1]   Jingrui He [1]

## Abstract

The Neural Tangent Kernel is a theoretical framework for understanding the training dynamics of neural networks. However, standard NTK and its variants fail to properly depict the finetuning of foundation models, as they neglect the preconditioning effects of adaptive gradients. To bridge this gap, we propose the Optimizer Aware Kernel (OAK), which incorporates the optimizer's adaptivity into standard NTK framework by a preconditioner estimation technique. Beyond the proposed method, we investigate a fundamental theoretical issue in the field: When and how the kernel regime collapses in finetuning. We derive explicit error bounds showing that the collapse of kernel regime is primarily due to the cumulative training effects and the task discrepancy between pretraining and finetuning. Theoretically, we justify OAK's preconditioner estimation by bounding its error term. Empirically, experiments on various model architectures show both the effectiveness of the OAK method and validity of our arguments on kernel regime collapse. Code is available at https://github.com/xiyuanyang45/Optimizer-Aware-Kernels.

## 1. Introduction

The Neural Tangent Kernel (NTK), first proposed by Jacot et al. (2018), provides a powerful theoretical framework for understanding the training dynamics of neural networks. It connects the gradient flow driven optimization to a deterministic kernel regression process (Jacot et al., 2018; Lee et al., 2018; Chizat & Bach, 2018). Specifically, in this kernel regime, the network's output changes linearly from its initial output, and the kernel matrix, which is defined by the inner product of gradients with respect to the model param-

eters, remains unchanged throughout training. With these approximations, we have the optimization of neural network the same as a kernel regression (Yang & Hu, 2020; Yang et al., 2022), where the features are gradients of samples.

The NTK has become a preliminary tool for modeling the training dynamics of various neural network architectures (Arora et al., 2019a; Ai et al., 2023; Malladi et al., 2023). Its empirical counterpart, the empirical NTK (eNTK), is useful for practical analysis. Unlike NTK defined in the infinite-width limit, the eNTK is computed for a finite-width network(Wei et al., 2022), which can be used in downstream tasks like predicting model's output, analyzing generalization performance, and understanding optimization landscapes (Mohamadi et al., 2023; Lisicki et al., 2021). With the rise of foundational models (Myers et al., 2024) (especially language models, e.g., Radford et al. (2019); Devlin et al. (2019)) based on the Transformer architecture, the NTK framework has been extended to analyze their finetuning dynamics. Malladi et al. (2023) shows that finetuning a pretrained model can also exhibit kernel-like behavior, where the pretrained parameters serve as the initialization point. Motivated by this, Malladi et al. (2023); Littwin & Yang (2023) have attempted to incorporate specific precondition effects to simulate adaptive optimizer behaviors.

Despite its progress, there is a critical gap between NTK and real optimization in modern foundation models: the standard NTK theory (Jacot et al., 2018) and its mentioned variants either mainly focus on the natural gradient flow or apply simple transformations on gradients. However, they neglect the core adaptivity (i.e., the coordinate-specific update scale) of modern optimizers, which is crucial in foundation model optimization. For example, existing mitigations like Malladi et al. (2023) make a heuristic based assertion that the SignSGD (Bernstein et al., 2018) works like Adam, and only consider the sign of each coordinate as features in NTK. This method enforces an identical update step size (feature scale) for each coordinate, and thus fails to adapt when certain parameters require vastly different update scales (i.e. exhibit different importance/significance to current sample). Furthermore, language model gradients often follow an extra heavy-tailed distribution (Yadav et al., 2023; Lee et al., 2025b), which consists of large gradient noise. Simply applying the sign function can amplify this noise to the same magnitude as informative gradient signals

---

[1]University of Illinois Urbana-Champaign. Correspondence to: Jingrui He <jingrui@illinois.edu>.

*Proceedings of the 43rd International Conference on Machine Learning*, Seoul, South Korea. PMLR 306, 2026. Copyright 2026 by the author(s).

(i.e., mapping both to $\pm1$), making it a noisy tool to model the true optimization process.

Previous works (Malladi et al., 2023) mainly focused on heuristic-based mitigations because they cannot directly calculate the corresponding preconditioner (which needs historical gradients) in a static kernel. This mismatch motivates our first research question: *1) How can we incorporate the preconditioning effect of adaptive optimizers into the NTK framework?* In this work, we propose the Optimizer Aware Kernel (OAK), using the finetuning dataset itself to create a static, one-time estimation of the optimizer's preconditioners (e.g., in Adam, we estimate the momentum (first moment) and variance (second moment)). These estimated vectors are then used to precondition the gradient features in NTK, thus faithfully simulating the optimizer's effect. Furthermore, we theoretically prove the effectiveness of this estimation by bounding its error to the real optimization process, which validates OAK as a theoretically sound extension of the NTK, rather than just another heuristic.

While OAK provides a faithful simulation of adaptive optimizers, it raises a more fundamental question about its underlying kernel regime. The core assumption of NTK is the fixed kernel matrix (i.e. fixed gradients), and a violation during finetuning can lead to a kernel collapse, where NTK-like methods do not work like finetuning anymore. The need of underlying kernel regime motivates our second question: *2) Under what conditions does the finetuning process deviate from the kernel regime, and what factors govern this deviation?* To answer this, we conduct a formal theoretical analysis of kernel behavior during finetuning, and our error bounds show that the regime deviation depends on several key factors, including cumulative training effects and the task discrepancy between pretraining and finetuning.

We summarize our contributions as follows:

- We propose the Optimizer Aware Kernel, a novel extension of the NTK framework designed for large language models with adaptive optimizers. OAK adds the preconditioning effect of adaptive optimizers by introducing the preconditioner estimation technique, providing a more faithful imitation of true training dynamics (Section 3).

- We theoretically explain when and how the NTK fails during finetuning of pretrained models. Our bounds show that the collapse of the kernel regime is a predictable function of the cumulative training step and the discrepancy between pretraining and finetuning tasks (Section 4). We also empirically validate our theoretical arguments by controlled experiments (Section 6.3).

- Theoretically, we prove the effectiveness of the OAK method by bounding the error term in our preconditioner estimation. We show that the error is small enough when the kernel regime holds (Section 5).

- Empirically, we conduct extensive experiments on a variety of natural language understanding tasks with various backbones. The results validate our theoretical findings and show that OAK consistently outperforms other strong baselines (Section 6).

## 2. Related Works

### 2.1. NTK and Kernel Regime

The NTK offers a theoretical tool for analyzing the training dynamics of highly overparameterized neural networks (Jacot et al., 2018; Arora et al., 2019a). The foundational work by Jacot et al. (2018) shows that a neural network optimized by gradient flow behaves as a linear model w.r.t. its initial parameters, and its evolution in the function space is governed by a static kernel—the NTK (Lee et al., 2019; Woodworth et al., 2020). This lazy training regime implies that network parameters barely move from their random initialization, and the model's predictions can be described by a first-order Taylor expansion around initial parameters.

However, the idealized lazy regime has limitations. In real world training, networks deviate from kernel behavior to the feature learning regime, where parameters change greatly and the NTK evolves dynamically (Yang & Hu, 2020). The degree to which a network operates in the lazy versus the feature learning regime is influenced by factors such as the depth-to-width ratio and initialization (Geiger et al., 2020; Hanin & Nica, 2019). However, these preliminary factors only hold on standard NTK frameworks with random LeCun initialization (Le et al., 2015). Additionally, the empirical NTK (Wei et al., 2022) has become an important tool to implement NTK and make predictions on a given model. Recently, Zeng et al. (2025) and Afzal et al. (2024) use the eNTK to analyze the finetuning dynamics of pretrained models, suggesting that this process also exhibits the kernel behavior. However, the performance gap between eNTK methods and finetuned models remains large according to their results (Malladi et al., 2023). Motivated by this, our work seeks to bridge this gap by providing a theoretical explanation for when and how the finetuning dynamics depart from the idealized lazy regime.

### 2.2. Adaptive Optimizer and Preconditioning

While standard NTK theory considers natural gradient flow, modern foundation models' training relies exclusively on adaptive gradients (e.g., Adam (Kingma, 2014)). Adaptive optimizers compute coordinate-specific step size by modifying raw gradients given the momentum and variance of historical gradients. From a theoretical view, this modification can be interpreted as applying a preconditioning matrix to the gradient vectors to perform gradient descent (Ye, 2024). The per-coordinate update is multiplied by the diagonal preconditioning matrix, which rescales coordinates to counteract a poorly conditioned loss landscape.

Adaptive optimizers outperform natural gradient optimization especially on large foundation models. Recent studies attribute this to the specific architectures and statistical properties of natural language tasks (Yadav et al., 2023). The natural gradient for these tasks exhibits more noise and is more heavy-tailed than in traditional architectures. This suggests that natural gradient flow of current NTK can be suboptimal in modeling the training dynamics of foundation models. To address this mismatch, we propose the OAK to incorporate the preconditioning effect of adaptive optimizers into the current NTK framework.

## 3. OAK with Preconditioner Estimation

To solve the question *1) How to incorporate the preconditioning effect of adaptive optimizers into the NTK framework?*, we introduce the Optimizer Aware Kernel as follows. We begin with the notation for the foundation model finetuning (Section 3.1). We then discuss the rationale for our core idea—the estimation of preconditioners (Section 3.2), and elaborate on the methodology for constructing the OAK matrix with pseudocode Algorithm 1 (Section 3.3). Finally, we discuss the compatibility of OAK with a range of adaptive optimizers, including both generally used Adam family and recent advances such as Muon (Section 3.4).

### 3.1. Notation and Preliminaries
We consider the task of finetuning a pretrained foundation model, represented by the function $f(\xi; \theta) : \mathbb{R}^{d_{in}} \to \mathbb{R}^o$, where $\xi$ is an input sample, $\theta \in \mathbb{R}^d$ are the model parameters, and $o$ is the output dimension. The finetuning process starts from the pretrained parameters $\theta_0$. We denote the pretraining and finetuning data distributions as $D_1$ and $D_2$, respectively. The objective is to minimize a loss function $\mathcal{L}(\xi; \theta) = l(f(\xi; \theta), y)$, where $l$ is a task-specific loss like cross-entropy or log-probability. The expected loss over the finetuning distribution is $\mathcal{L}_{FT}(\theta) = \mathbb{E}_{(\xi,y) \sim D_2}[\mathcal{L}(\xi; \theta)]$.

The NTK framework approximates the model's evolution through its first-order Taylor expansion around $\theta_0$. The key components are the per-sample gradient features, also known as Jacobians, defined as $J(\xi; \theta) = \nabla_\theta f(\xi; \theta)$. The standard empirical NTK matrix is then constructed as the inner product of these initial gradient features: $K_{\text{NTK}}(\xi_a, \xi_b) = \langle \nabla_\theta f(\xi_a; \theta), \nabla_\theta f(\xi_b; \theta) \rangle$. Here the Jacobian follows the numerator layout $J(\xi; \theta) = \nabla_\theta f(\xi; \theta) \in \mathbb{R}^{o \times d}$, and for multi-output models ($o > 1$) the inner product $\langle \cdot, \cdot \rangle$ denotes the Frobenius inner product $\langle J_a, J_b \rangle = \text{Tr}(J_a J_b^T)$, which yields a scalar kernel entry and is implemented by the trace operation in Algorithm 1. We note that this formulation for NTK is derived from gradient descent (or natural gradient flow), which, as we have argued, fails to capture the preconditioning effect of adaptive optimizers in the finetuning process of foundation models.

### 3.2. Rationale for Preconditioner Estimation
A primary challenge of incorporating adaptive effects into NTK is the dynamic preconditioners. For instance, the preconditioner of Adam ($m_t$ and $v_t$) change over time, thus cannot be directly added into the static kernel. This is also the reason why previous works cannot consider the adaptivity. In this work, we aim to derive a static estimation of preconditioners. Before we go into method details, we list the rationale of this estimation (why it can be estimated) as follows through discussion over several literatures:

- First, preconditioners exhibit sample-level stability. For example, Li et al. (2022) has shown that preconditioners can be effectively estimated using relative data, and are not overly sensitive to individual samples. Our approach leverages the finetuning set in a similar manner to estimate a robust preconditioner.

- Second, preconditioners exhibit temporal stability. For instance, Li et al. (2023) shows that aggregating noisy preconditioners over steps yields a more stable estimate. This stability implies that a static estimation can also serve as a reasonable proxy.

- Finally, the exact value of each coordinate in the preconditioner is less important than its overall scaling effect. Lee et al. (2025a) found that the Adam's preconditioning can be approximated by a simple BiClip operator that directly clamps the gradient coordinates within a range.

According to these observations, it is reasonable to say that "preconditioner is relatively stable, and can be estimated by relative information". Therefore, we propose OAK, a principled method to incorporate adaptive effects into the NTK, as follows.

### 3.3. Proposed OAK Method
In this section, we elaborate on how to construct the kernel matrix of OAK given the finetune set ($S_{\text{train}} = \{(\xi_i, y_i)\}_{i=1}^{N_{\text{train}}}$) and the test set ($S_{\text{test}} = \{\xi_j\}_{j=1}^{N_{\text{test}}}$), which can be used in downstream tasks such as test set classification. The construction of the OAK matrix (Algorithm 1) follows a three-step process: i) estimating the preconditioners of the adaptive optimizer (here we use Adam as an example), ii) preconditioning the gradient features of the target samples, and iii) computing the kernel matrix.

First, given a pretrained model $f$ with parameters $\theta_0$ and the finetuning (training) set $S_{\text{train}} = \{(\xi_i, y_i)\}_{i=1}^{N_{\text{train}}}$, we first calculate the estimated momentum $\hat{m}_0$ and variance $\hat{v}_0$. We compute the per-sample loss gradients over this set and average them as prescribed in the `PreconditionerEstimation` function:

$$\hat{m}_0 = \frac{1}{N_{\text{train}}} \sum_{i=1}^{N_{\text{train}}} g_i, \quad \hat{v}_0 = \frac{1}{N_{\text{train}}} \sum_{i=1}^{N_{\text{train}}} g_i \odot g_i.$$

---

**Algorithm 1** OAK (Adam) as an example

---

**Input:** Pretrained model $f$ with parameters $\theta_0$; Finetuning set $S_{\text{train}} = \{(\xi_i, y_i)\}_{i=1}^{N_{\text{train}}}$; Target (test) set $S_{\text{test}} = \{\xi_j\}_{j=1}^{N_{\text{test}}}$; Adam hyperparameters $\beta_1, \beta_2, \epsilon$.

**Output:** OAK matrix $K \in \mathbb{R}^{N_{\text{train}} \times N_{\text{test}}}$.

1: **func** OAK_Kernel($f, \theta_0, S_{\text{train}}, S_{\text{test}}$)
2:      $\hat{m}_0, \hat{v}_0 \leftarrow$ Prec.Estimation($f, \theta_0, S_{\text{train}}$)
3:      $J_i \leftarrow \nabla_\theta f(\xi_i; \theta_0), \forall \xi_i \in S_{\text{train}}$
4:      $J'_j \leftarrow \nabla_\theta f(\xi_j; \theta_0), \forall \xi_j \in S_{\text{test}}$
5:      $\tilde{J}_j \leftarrow$ AdamPrec.($J'_j, \hat{m}_0, \hat{v}_0$), $\forall j \in [N_{\text{test}}]$
6:      $K_{ij} \leftarrow \text{Tr}(J_i \tilde{J}_j^T), \forall i \in [N_{\text{train}}]$ and $\forall j \in [N_{\text{test}}]$
7:      **return** $K$
8: **end func**
9: **func** PreconditionerEstimation($f, \theta_0, S_{\text{train}}$)
10:      $g_i \leftarrow \nabla_\theta \mathcal{L}(f(\xi_i; \theta_0), y_i), \forall (\xi_i, y_i) \in S_{\text{train}}$.
11:      $\hat{m} = \frac{1}{N_{\text{train}}} \sum_{i=1}^{N_{\text{train}}} g_i$; $\hat{v} = \frac{1}{N_{\text{train}}} \sum_{i=1}^{N_{\text{train}}} g_i \odot g_i$
12:      **return** $\hat{m}, \hat{v}$
13: **end func**
14: **func** AdamPreconditioning($J, \hat{m}_0, \hat{v}_0, \beta_1, \beta_2, \epsilon$)
15:      $m' \leftarrow \beta_1 \hat{m}_0 + (1 - \beta_1)J$
16:      $v' \leftarrow \beta_2 \hat{v}_0 + (1 - \beta_2)J \odot J$
17:      **return** $m' \oslash (\sqrt{v'} + \epsilon)$
18: **end func**

---

where $g_i$ is the model's gradient on the $i$-th sample in $S_{\text{train}}$, and $\odot$ denotes the element-wise product. This procedure (Algorithm 1, Line 2) provides a global, data-driven estimate of the optimizer's initial state.

Next, for the input samples in the training set $S_{\text{train}}$ and the target (test) set $S_{\text{test}}$, we compute their respective Jacobians, $J_i$ and $J'_j$ (Algorithm 1, Lines 3-4). The core of our method lies in transforming the test set gradients using our estimated preconditioners. As shown in AdamPreconditioning in Algorithm 1, each test set Jacobian $J'_j$ is transformed into its preconditioned version $\tilde{J}_j$ by applying a procedure analogous to a single Adam update step:

$$\tilde{J}_j = \frac{\beta_1 \hat{m}_0 + (1 - \beta_1)J'_j}{\sqrt{\beta_2 \hat{v}_0 + (1 - \beta_2)J'_j \odot J'_j} + \epsilon}.$$

Here, the estimated $\hat{m}_0$ and $\hat{v}_0$ (from $S_{\text{train}}$) are the moments from the previous step ($m_{t-1}, v_{t-1}$), while each test set Jacobian $J'_j$ acts as the current gradient ($g_t$). This step (Algorithm 1, Line 5) directly follows the way that adaptive optimizers rescale the feature space for the target samples.

Finally, the OAK matrix $K \in \mathbb{R}^{N_{\text{train}} \times N_{\text{test}}}$ is constructed by computing the inner product between the unmodified Jacobians from the training set, $J_i$, and the preconditioned Jacobians from the test set, $\tilde{J}_j$ (Algorithm 1, Line 6). This construction faithfully models adaptive optimization, where the change in the model's output for a test sample is determined by the inner product of the original gradient features

and the preconditioned update direction. As a result, the obtained OAK matrix $K$ can then be directly used within a standard kernel regression framework to make further predictions for the target set or on downstream tasks.

**Training Gram Matrix and Kernel Regression.** For the implementation of kernel regression with training and test data, the training Gram matrix $K_{\text{train}} \in \mathbb{R}^{N_{\text{train}} \times N_{\text{train}}}$ is constructed with the same preconditioning mechanism: for any $\xi_i, \xi_{i'} \in S_{\text{train}}$, $K_{\text{train}}(i, i') = \text{Tr}(J_i \tilde{J}_{i'}^T)$, where $J_i$ is the raw Jacobian of $\xi_i$ and $\tilde{J}_{i'}$ the preconditioned Jacobian of $\xi_{i'}$. The target predictions are then $\hat{y}_j = \sum_{i=1}^{N_{\text{train}}} \alpha_i K_{ij}$ with $\alpha = K_{\text{train}}^{-1} y_{\text{train}}$, where $K_{ij}$ is the OAK matrix from Algorithm 1 and $y_{\text{train}}$ are the training labels.

### 3.4. Discussion on OAK Compatibility

While we use Adam as the example, the OAK is compatible with other adaptive optimizers. For optimizers like **RMSProp** (Tieleman & Hinton, 2012) and **AdaGrad** (Duchi et al., 2011), which primarily rely on the variance (second moment) of gradients, the preconditioning step can fallback to only using the estimated $\hat{v}_0$ from our PreconditionerEstimation function, and the preconditioned Jacobian $\tilde{J}_j$ would be computed as $J'_j \oslash (\sqrt{\hat{v}_0} + \epsilon)$, where $\oslash$ is the element-wise division. To adapt OAK for **AdamW** (Loshchilov & Hutter, 2019), which introduces a decoupled weight decay, we can explicitly incorporate the weight decay term into the preconditioned features. After computing the Adam-preconditioned Jacobian $\tilde{J}_j$ as detailed in our main methodology, the final feature vector used in the kernel construction would become $(\tilde{J}_j + \lambda \theta_0)$, where $\lambda$ is the weight decay coefficient. This adaptability demonstrates that OAK provides a generalizable framework for incorporating the effects of various adaptive optimizers into the standard NTK framework. We conduct experiments on incorporating different optimizers for OAK in Section 6.2.

More broadly, OAK injects an optimizer-specific transformation $\mathcal{T}(\cdot)$ on the gradient features and is not restricted to diagonal preconditioners. Beyond the Adam family, OAK directly accommodates matrix-based optimizers such as **Muon** (Liu et al., 2025) or **MuonClip** (Kimi Team, 2025): we instantiate $\mathcal{T}$ as the per-layer Newton–Schulz orthogonalization $\mathcal{T}(M) = \text{NS}(M) \approx UV^T$ applied to the (moment-augmented) layer Jacobian.

## 4. Kernel Behavior Analysis

In this section, we theoretically analyze the dynamics of finetuning to determine when and how the model's behavior deviates from the kernel regime. For simplicity, we consider the network output as a scalar ($o = 1$, taken as the logit of the ground-truth class), following standard NTK practice (Jacot et al., 2018; Malladi et al., 2023); this keeps the model Hessian $H_f(\theta) \in \mathbb{R}^{d \times d}$ well-defined and avoids

third-order tensors. The multi-output construction of Section 3.3 reduces to this case entry-wise via the Frobenius inner product. Following the standard NTK theorem (Jacot et al., 2018), the kernel regime is characterized by two key properties: linearization, where the model's output behaves like its first-order Taylor expansion, and fixed features, where the model's gradients remain constant. We formally define these properties below.

**Definition 4.1** (Linearization). The finetuning process exhibits linearization if the model function at any step $t$, $f(\xi; \theta_t)$, remains close to its first-order Taylor expansion around the initial parameters $\theta_0$ as: $||f(\xi; \theta_t) - f(\xi; \theta_0)||_2 \approx \langle \nabla_\theta f(\xi; \theta_0)^T, \theta_t - \theta_0 \rangle$.

**Definition 4.2** (Fixed Features). The finetuning process exhibits fixed features if at any step $t$, the feature vector $\nabla_\theta f(\xi; \theta)$ (gradient) remains unchanged throughout training, as: $\nabla_\theta f(\xi; \theta_0) \approx \nabla_\theta f(\xi; \theta_t)$.

Our analysis aims to answer question *2)* from the following two aspects: *2.a) Under what conditions can the kernel regime mild fail during finetuning?* and *2.b) Is this failure a gradual fail or a catastrophic breakdown?*. To build our analysis, we need additional notation (Section 4.1) and assumptions (Section 4.2) about finetuning. Then we derive lemmas that bound the initial gradient and curvature (Section 4.3), which are crucial for following theorems (Section 4.4). Due to page limits, we put the formal proof of Lemma 4.8-4.9 and Theorem 4.10-4.11 in Appendix B.1-B.4.

### 4.1. Additional Notation

Following the finetuning setting and notation in Section 3.1, we add additional notation as follows: We denote the pretraining loss as $\mathcal{L}_{PT}(\theta) = \mathbb{E}_{(\xi,y)\sim D_1}[\mathcal{L}(\xi; \theta)] = \mathbb{E}_{(\xi,y)\sim D_1}[l(f(\xi; \theta), y)]$, and the finetuning loss as $\mathcal{L}_{FT}(\theta) = \mathbb{E}_{(\xi,y)\sim D_2}[\mathcal{L}(\xi; \theta)] = \mathbb{E}_{(\xi,y)\sim D_2}[l(f(\xi; \theta), y)]$. The minimizers (or one of all minimizers, can be local minima) of $\mathcal{L}_{PT}$ and $\mathcal{L}_{FT}$ are $\theta_{PT}^*$ and $\theta_{FT}^*$ respectively. Since pretraining process is often sufficient for model parameters to fit the underlying distribution $D_1$, we have that $\theta_0$ is an approximation of $\theta_{PT}^*$, such that $||\theta_0 - \theta_{PT}^*||_2 \leq \delta_{PT}$, where $\delta_{PT}$ is the pretraining optimization error. We also denote the Hessian Matrix of the model function as $H_f(\theta) = \nabla_\theta^2 f(\xi; \theta) \in \mathbb{R}^{d \times d}$.

### 4.2. Assumptions

**Assumption 4.3** (Lipschitz Continuous Gradient w.r.t. Parameters). During finetuning of a pretrained model, for any model parameter $\theta_a, \theta_b \in \mathbb{R}^d$, and input $\xi$, we have $||\nabla \mathcal{L}(\xi; \theta_a) - \nabla \mathcal{L}(\xi; \theta_b)||_2 \leq L_\theta ||\theta_a - \theta_b||_2$.

**Assumption 4.4** (Lipschitz Continuous Gradient w.r.t. Sample). During finetuning of a pretrained model, for any sample $\xi_a, \xi_b$, and model parameter $\theta$, we have $||\nabla \mathcal{L}(\xi_a; \theta) - \nabla \mathcal{L}(\xi_b; \theta)||_2 \leq L_\xi ||\xi_a - \xi_b||_2$.

**Assumption 4.5** (Polyak-Lojasiewicz Condition). The finetuning loss $\mathcal{L}_{FT}(\theta)$ satisfies the $\mu$-PL condition. This means

the distance from any parameter $\theta_t$ to the loss minimizer $\theta_{FT}^*$ is bounded by: $||\theta_t - \theta_{FT}^*||_2 \leq \frac{1}{\mu}||\nabla \mathcal{L}_{FT}(\theta_t) - \nabla \mathcal{L}_{FT}(\theta_{FT}^*)||_2$.

**Assumption 4.6** (Lipschitz Continuous Hessian). The model's Hessian matrix, $H_f(\theta)$, is $L_H$-Lipschitz continuous: $||H_f(\theta_a) - H_f(\theta_b)||_2 \leq L_H ||\theta_a - \theta_b||_2$.

**Assumption 4.7** (Bounded Loss Curvature). We assume the (spectral) norm of the Hessian matrix on $\theta_{PT}^*$ is upper bounded: $||H_f(\theta_{PT}^*)||_2 \leq C_{PT}$, where $C_{PT}$ is a constant.

Assumptions 4.3, 4.4, and 4.6 are common conditions that hold locally for models with smooth activation (Bottou et al., 2018; Reddi et al., 2019). The Polyak-Lojasiewicz condition (Assumption 4.5) is a widely used relaxation of strong convexity that has been observed to hold in overparameterized models (Karimi et al., 2016). Finally, Assumption 4.7 is motivated by empirical observations that pretraining with stochastic optimizers tends to find solutions in flat regions of the loss landscape, which correspond to low-norm Hessians (Keskar et al., 2016; Chaudhari et al., 2019).

### 4.3. Lemmas

Based on these assumptions, we derive two key lemmas that provide the foundation for our main theorems. Lemma 4.8 formalizes the intuition that the gradient for finetuning is driven by the discrepancy between the pretraining and finetuning data distributions. Lemma 4.9 establishes that the model's curvature at the start of finetuning is well-behaved, inheriting the same curvature property from the pretrained solution.

**Lemma 4.8** (Bounding Finetuning Gradient at $\theta_0$). *With Assumptions 4.3-4.5, the gradient norm (of fine-tuning loss) at the initial parameter point $\theta_0$ is bounded by $\epsilon_g$:*

$$||\nabla \mathcal{L}_{FT}(\theta_0)||_2 \leq \epsilon_g \tag{1}$$

$$where \quad \epsilon_g \leq L_\theta \left( \delta_{PT} + \frac{L_\xi}{\mu} W_1(D_1, D_2) \right). \tag{2}$$

*where $W_1(D_1, D_2)$ is the Wasserstein-1 distance between the pretraining data distribution $D_1$ and the fine-tuning data distribution $D_2$.*

**Lemma 4.9** (Bounding Model Hessian at $\theta_0$). *With Assumptions 4.6 and 4.7, the norm of the model function's Hessian at pretrained parameters $\theta_0$ is bounded by $\epsilon_f$:*

$$||H_f(\theta_0)||_2 \leq \epsilon_f \quad where \quad \epsilon_f \leq L_H \cdot \delta_{PT} + C_{PT}. \tag{3}$$

### 4.4. Main Results

To quantify how closely the finetuning process adheres to the kernel regime, we now derive explicit bounds on the error terms for the linearization (Definition 4.1) and fixed features (Definition 4.2). These bounds reveal how the properties of the pretrained model, the finetuning task, and the optimization process jointly affect whether the kernel regime still holds as follows.

**Theorem 4.10** (Error Term in Linearization). *With Assumptions 4.3-4.7, given finetuning for $T$ steps with learning rate $\eta$ such that $T\eta L_\theta < 1$, the linearization error, defined as the Lagrange remainder term $R = f(\xi; \theta_T) - (f(\xi; \theta_0) + \langle \nabla_\theta f(\xi; \theta_0), \theta_T - \theta_0 \rangle)$, is bounded by:*

$$||R||_2 \leq \frac{1}{2} \left( \frac{T\eta\epsilon_g}{1 - T\eta L_\theta} \right)^2 \left( \epsilon_f + L_H \frac{T\eta\epsilon_g}{1 - T\eta L_\theta} \right), \quad (4)$$

*where $\epsilon_g$ and $\epsilon_f$ are the upper-bounds of the initial finetuning gradient and model Hessian norm, respectively.*

**Theorem 4.11** (Error Term in Fixed Features). *With Assumptions 4.3-4.7, given finetuning for $T$ steps with learning rate $\eta$ such that $T\eta L_\theta < 1$, the feature drift, defined as the deviation of the gradient features at step $T$ from their initial state $E = ||\nabla_\theta f(\xi; \theta_T) - \nabla_\theta f(\xi; \theta_0)||_2$, is bounded by:*

$$||E||_2 \leq \left( \epsilon_f + L_H \frac{T\eta\epsilon_g}{1 - T\eta L_\theta} \right) \left( \frac{T\eta\epsilon_g}{1 - T\eta L_\theta} \right), \quad (5)$$

*where $\epsilon_g$ and $\epsilon_f$ are the upper-bounds of the initial finetuning gradient and model Hessian norm, respectively.*

**Remark 4.12** (Extension to Adaptive Optimizers). *Theorems 4.10–4.11 are stated for raw gradient, but extend to adaptive optimizers (like Adam) as follows: For a diagonal preconditioner $P_t$ with eigenvalues in $[c_1, c_2]$, the update satisfies $||\Delta\theta_t||_2 \leq c_2\eta||\nabla\mathcal{L}(\theta_t)||_2$. Repeating Eq. (26) under the condition $c_2 T\eta L_\theta < 1$ gives $D_T^{Adam} \leq c_2 D_T$. The error bounds are thus inflated by constants in $c_2$ (to leading order, $||R||_2^{Adam} \lesssim c_2^2 ||R||_2$, $||E||_2^{Adam} \lesssim c_2 ||E||_2$), leaving the dependence on $T\eta$, $\epsilon_g$, and task discrepancy unchanged.*

### 4.5. Discussion on Results

The bounds derived in Theorem 4.10 and Theorem 4.11 provide a quantitative basis for understanding whether the kernel behavior holds during finetuning. Both error bounds share a common structure, and their magnitudes are commonly related to few key factors:

- **Cumulative Training Step** ($T\eta$)**:** This is the most critical factor in both bounds. Both bounds are highly sensitive to the term $(1 - T\eta L_\theta)$ in the denominator. As finetuning steps $T$ and learning rate $\eta$ increase, both the linearization error and feature drift to increase rapidly.

- **Task Discrepancy** ($\epsilon_g$)**:** The initial gradient norm, $\epsilon_g$, which is bounded by the Wasserstein distance $W_1(D_1, D_2)$ between pretraining and finetuning data distributions, acts as the main driving force for parameter movement. A larger discrepancy between tasks can lead to a larger initial gradient, and finally results in a faster kernel behavior collapse.

- **Model Curvature** ($\epsilon_f$ and $L_H$)**:** The initial Hessian norm $\epsilon_f$ and its Lipschitz constant $L_H$ characterize the curvature of the loss landscape. A highly curved or rapidly changing landscape (large $\epsilon_f$ or $L_H$) amplifies the non-linear effects, leading to a larger linearization error and feature shift.

With these results, we answer the questions as follows:

**2.a) Kernel behavior fails as training progresses.** Our bounds show that both linearization error and feature drift grow with the cumulative training step $T\eta$. Therefore, for any non-trivial finetuning task where $\epsilon_g > 0$, the model will eventually deviate from the kernel regime. The NTK approximation is only valid in the early stages of finetuning.

**2.b) The failure transitions from mild to catastrophic.** The failure depends on the magnitude of $T\eta$. When $T\eta \ll 1/L_\theta$, the error bounds are very small. This corresponds to an initial **mild failure**, where the training dynamics can be approximated by a linear model, and the features (gradients) remain stable. However, as $T\eta$ (especially $T$) grows, the bounds approach infinity. This signals a **catastrophic failure** of the kernel approximation, representing a fast deviation from the lazy training regime.

## 5. Preconditioner Estimation Analysis

In this section, we theoretically justify our preconditoner estimation by bounding its error. We show that this error is small within the kernel regime, thereby justifying our approach. The analysis is composed of addressing the two components of the Adam preconditioner: the momentum and variance. We put the formal proof of Theorem 5.2 and 5.4 in Appendix B.5 and B.6 respectively.

### 5.1. Momentum Estimation

We begin by analyzing the error in the momentum estimation. We first introduce a standard assumption on gradient variance.

**Assumption 5.1** (Bounded Gradient Variance). The variance of the per-sample gradients with respect to the finetuning data distribution $D_2$ is bounded by $\sigma^2$ at the initial parameter point $\theta_0$:

$$\mathbb{E}_{(\xi,y)\sim D_2}[||\nabla\mathcal{L}(\xi; \theta_0) - \nabla\mathcal{L}_{FT}(\theta_0)||_2^2] \leq \sigma^2. \quad (6)$$

With this assumption, we can now bound the total error of our estimation $\hat{m}_0 = \frac{1}{N}\sum_{i=1}^N \nabla\mathcal{L}(\xi_i, y_i; \theta_0)$ on the true momentum $m_T$.

**Theorem 5.2** (Momentum Estimation Error Bound). *With Assumptions 4.3-5.1, let $m_T$ be the true momentum vector after $T$ steps of Adam optimization with hyperparameter $\beta_1$ and learning rate $\eta$. We have with probability at least $(1-\delta)$, the error of the estimated momentum $\hat{m}_0$ is bounded by:*

$$||\hat{m}_0 - m_T||_2 \leq \underbrace{\frac{\sigma}{\sqrt{N\delta}}}_{\text{Sampling Error}}$$
$$+ \underbrace{(1 - \beta_1^T)L_\theta D_T + \beta_1^T \epsilon_g}_{\text{Dynamic Error}}, \quad (7)$$

*where $D_T = \frac{T\eta\epsilon_g}{1 - T\eta L_\theta}$ is the parameter drift bound from the proof of Theorem 4.10 (see Appendix B.3).*

## 5.2. Variance Estimation

We now extend the analysis to the second moment. Our estimate is $\hat{v}_0 = \frac{1}{N} \sum_{i=1}^{N} \nabla \mathcal{L}(\xi_i, y_i; \theta_0)^2$. To bound its error relative to true variance $v_T$, we require an assumption on the higher-order moments of the gradient.

**Assumption 5.3** (Bounded Moment of Gradient). To bound the sampling error of the variance, we assume the fourth moment of the per-sample gradient norm is bounded at $\theta_0$ by a constant $M$:

$$\mathbb{E}_{(\xi,y) \sim D_2}[||\nabla \mathcal{L}(\xi; \theta_0)||_2^4] \le M. \tag{8}$$

**Theorem 5.4** (Variance Estimation Error Bound). *With Assumptions 4.3-5.3, let $v_T$ be the true variance vector after $T$ steps of Adam with parameter $\beta_2$. We have with probability at least $(1 - \delta)$, the error of the estimated second moment $\hat{v}_0$ is bounded by:*

$$||\hat{v}_0 - v_T||_2 \le \underbrace{\sqrt{\frac{M - (\sigma^2 + \epsilon_g^2)^2}{\delta N}}}_{\text{Sampling Error}}$$
$$+ \underbrace{\begin{array}{c} (1 - \beta_2^T) \left(2\epsilon_g L_\theta D_T + (L_\theta D_T)^2\right) \\ + \beta_2^T (\sigma^2 + \epsilon_g^2) \end{array}}_{\text{Dynamic Error}}, \tag{9}$$

*where $D_T = \frac{T \eta \epsilon_g}{1 - T \eta L_\theta}$ is the parameter drift bound from the proof of Theorem 4.10 (see Appendix B.3).*

## 5.3. Discussions on Error Bounds

Theorem 5.2 and Theorem 5.4 show that the validity of OAK's static preconditioner estimation is governed by the same factors that determine the kernel regime.

The error in our estimations is decomposed into two parts. The **Sampling Error** can be controlled and made arbitrarily small by increasing the size $N$ of the estimation set $S_{\text{est}}$ (for which we often use the whole finetuning set). The more significant component is the **Dynamic Error**, which quantifies how the true Adam moments drift from their initial state. This error is a direct function of the parameter drift $D_T$, which in turn is driven by the cumulative training step $(T\eta)$, the task discrepancy in $\epsilon_g$, and the model's curvature ($L_\theta$). This implies that OAK provides more accurate approximation of the optimizer aware kernel in the regime where the standard NTK is most relevant. Therefore, OAK serves as a principled extension of the NTK, offering a more accurate imitation of actual training dynamics of adaptive optimizers.

## 6. Experimental Results

In this section, we conduct a series of experiments to empirically validate our claims. Our primary objectives are as follows. First, in Section 6.2, we demonstrate the effectiveness of our proposed OAK by comparing it against

several baseline NTK methods on standard tasks for foundational models. We also verify the compatibility of the OAK framework with other common adaptive optimizers beyond Adam, and present a hyperparameter analysis. Second, in Section 6.3, we empirically validate our theoretical findings from Section 4. We systematically control for factors including training progress and task discrepancy to demonstrate the conditions that kernel regime collapses in finetuning.

## 6.1. Experimental Settings

**Model Architectures.** Our experiments focus on the finetuning scenario for modern foundation model backbones with two architectures: **encoder-based models** (RoBERTa (Liu et al., 2019)) (Radford et al., 2019), which are used to learn data representations; **decoder-only autoregressive models** with causal masking (Qwen2.5 (Yang et al., 2024)), which are used for next-token prediction. For all kernel-based methods, we perform the empirical NTK on finetune dataset and perform kernel regression to get predictions. All results are average accuracy of five trials.

**Tasks.** We evaluate all methods on the standard GLUE benchmark (Wang et al., 2018). We select a representative subset including single-sentence classification (SST-2 and COLA) and sentence-pair classification tasks (MNLI, QQP and MRPC). In our result tables, $n$ denotes the number of samples in the finetuning set, which is randomly selected from the official training split for each task.

**Hyperparameters.** For our OAK method, the hyperparameters are related to the specific adaptive optimizer being modeled. For Adam-based variants, we perform a grid search over the momentum $\beta_1 \in \{0.95, 0.99\}$ and the variance $\beta_2 \in \{0.9, 0.99, 0.999\}$. For the Adam finetuning baseline (Adam FT), we use a standard learning rate of $2 \times 10^{-5}$ and select the best model checkpoint across 1 to 20 finetuning epochs.

**Baselines.** We mainly include two baselines for comparison. For NTK, we directly perform the standard empirical NTK on the given pretrained model. SignGD-NTK is the method from Malladi et al. (2023), which used SignGD based transformation on gradient features as a heuristics based Adam approximation.

## 6.2. Effectiveness of OAK

**Performance Evaluation.** As shown in Tables 1,3,6 and 2, OAK achieves the highest accuracy in kernel methods, and even outperform the full finetuning in low-data regimes ($n = 8, 16$). This phenomenon is consistent with previous work (Arora et al., 2019b) which shows NTK-based methods can be effective on small-data tasks. The rationale is that directly finetuning a large model on a few examples may lead to overfitting, while NTK's kernel regression provides implicit regularization. The overall superior performance validates the effectiveness of OAK. Due to the page limit,

*Table 1.* Accuracy with RoBERTa on Double-Sentence Tasks. For NTK methods, we highlight the best and second-best score in **bold** and underlining. See Section 6.2 Performance Evaluation, full results in Appendix A.1.

| Method | MNLI | | | | MRPC | | | | QQP | | | | Avg. Acc. |
|---|---|---|---|---|---|---|---|---|---|---|---|---|---|
| | $n=8$ | $n=16$ | $n=32$ | $n=64$ | $n=8$ | $n=16$ | $n=32$ | $n=64$ | $n=8$ | $n=16$ | $n=32$ | $n=64$ | |
| Finetune | 33.46 | 33.95 | 34.44 | 35.48 | 68.38 | 70.26 | 70.87 | 73.15 | 53.65 | 54.77 | 61.26 | 61.57 | 54.27 |
| NTK | 33.46 | 33.85 | 33.53 | 34.11 | 55.75 | 54.66 | 56.95 | 61.76 | 47.20 | 49.48 | 53.45 | 54.33 | 47.38 |
| SignGD-NTK | 32.42 | 36.00 | 36.02 | 37.57 | 61.52 | 61.76 | 63.77 | **65.01** | 53.16 | 57.62 | 60.61 | 60.95 | 52.20 |
| OAK (Adam) | **33.85** | **36.96** | **36.07** | **37.66** | **62.50** | **62.83** | **64.62** | 62.85 | **54.69** | **59.24** | **61.72** | **61.87** | **52.91** |

*Table 2.* Accuracy with Qwen2.5-0.5B backbone on Double-Sentence Tasks. For NTK methods, we highlight the best and second-best score with **bold** and underlining. See Section 6.2 Performance Evaluation, full results in Appendix A.1.

| Method | MNLI | | | | MRPC | | | | QQP | | | | Avg. Acc. |
|---|---|---|---|---|---|---|---|---|---|---|---|---|---|
| | $n=8$ | $n=16$ | $n=32$ | $n=64$ | $n=8$ | $n=16$ | $n=32$ | $n=64$ | $n=8$ | $n=16$ | $n=32$ | $n=64$ | |
| Finetune | 31.97 | 34.18 | 36.78 | 44.01 | 53.89 | 54.17 | 54.62 | 58.99 | 45.51 | 56.97 | 57.39 | 64.65 | 49.43 |
| NTK | 29.36 | 33.72 | 33.92 | 33.01 | **54.25** | 52.70 | 54.98 | 54.08 | 51.56 | **55.99** | 55.66 | 56.30 | 46.30 |
| SignGD-NTK | 30.14 | 33.07 | 32.23 | 34.24 | 53.02 | 52.29 | 54.58 | 54.82 | 53.52 | 54.10 | 55.66 | 53.78 | 46.79 |
| OAK (Adam) | **30.86** | **33.94** | **34.07** | **34.38** | 52.86 | **53.47** | **55.23** | **55.64** | **54.88** | 55.95 | **58.46** | **59.67** | **48.28** |

we put Table 6 in the Appendix.

**Compatibility of OAK.** We want to note that OAK can also model other adaptive optimizers, as discussed in Section 3.4. Table 4 shows the compatibility results on the MNLI task ($n$=64). We observe that OAK can also work on AdaGrad and AdamW with strong performance. This shows our core idea of preconditioner estimation is a generalizable principle for incorporating various adaptive effects.

**Ablation Study.** The Adam optimizer can be seen as combining momentum (from SGDM (Sutskever et al., 2013)) and variance (from AdaGrad) together. We conduct an ablation study in Table 5 by implementing OAK with only one of these components. The results that OAK (Adam) outperforms both OAK (SGDM) and OAK (AdaGrad), indicates both the momentum and the variance preconditioning components are effective and crucial for modeling finetuning.

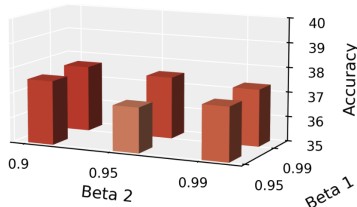

*Figure 1.* Hyperparameter sensitivity analysis for OAK. See Section 6.2 Hyperparameter Analysis.

**Hyperparameter Analysis.** We want to note that OAK itself does not have any hyperparameter. The only hyperparameter depends on the used optimizer. In Figure 1, we analyze the sensitivity of OAK (Adam) to choices of $\beta_1$ and $\beta_2$. The results show that OAK's performance is generally over 37%, and is not overly sensitive on hyperparameters.

**Computational Cost and Broader Applicability.** We further verify OAK's practicality from three aspects as follows: First, OAK shares the same asymptotic complexity as standard NTK and adds at most ~10% empirical overhead even

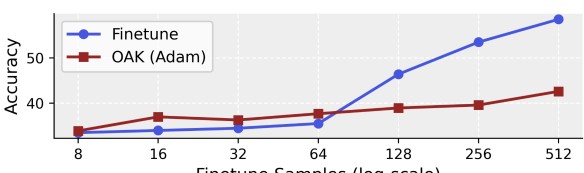

*Figure 2.* Performance of finetuning and OAK as training step increases. See Section 6.3.

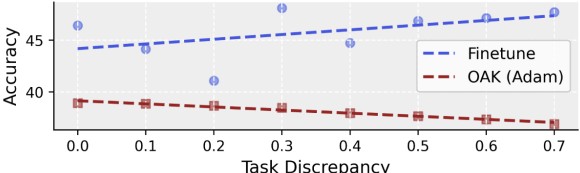

*Figure 3.* Performance of finetuning and OAK as task discrepancy (shuffle ratio) increases. See Section 6.3.

on 1.5B models (Appendix A.2). Second, beyond classification, OAK remains effective on single-step next-token generation (Appendix A.3). Third, OAK provides practical utility for downstream data selection, where subsets selected by OAK match full-data finetuning (Appendix A.4).

### 6.3. Validation of Kernel Behavior Analysis

Our theoretical analysis in Section 4 predicts that the kernel regime will fail as a function of cumulative training steps ($T\eta$) and task discrepancy ($\epsilon_g$). We now validate our arguments empirically as follows.

**Kernel Regime Collapses as Training Progresses.** Theorems 4.10 and 4.11 predict that kernel regime collapses as training steps $T\eta$ increase. Figure 2 empirically validates this: as the sample size $n$ grows (serving as a proxy for $T\eta$), the performance gap widens significantly, confirming the theoretical breakdown of the kernel approximation.

**Kernel Regime Collapses with Greater Task Discrepancy.** Our theory shows that a discrepancy in pretraining distribution ($D_1$) and finetuning distribution ($D_2$) can also

*Table 3.* Accuracy with RoBERTa backbone on Single-Sentence Tasks. For NTK methods, we highlight the best and second-best score with **bold** and underlining. See Section 6.2 Performance Evaluation, full results in Appendix A.1.

| Method | SST2 | | | | COLA | | | | Avg. Acc. |
|---|---|---|---|---|---|---|---|---|---|
| | $n = 8$ | $n = 16$ | $n = 32$ | $n = 64$ | $n = 8$ | $n = 16$ | $n = 32$ | $n = 64$ | |
| Finetune | 52.28 | 57.33 | 61.26 | 74.06 | 57.99 | 59.93 | 61.58 | 64.58 | 61.13 |
| NTK | 48.70 | 49.87 | 49.79 | 50.59 | 50.13 | 50.67 | 54.23 | 55.84 | 51.23 |
| SignGD-NTK | 53.76 | 58.20 | 60.61 | 67.51 | 54.56 | 58.59 | 60.03 | 60.12 | 59.17 |
| OAK (Adam) | **55.60** | **59.51** | **65.89** | **68.95** | **55.86** | **60.81** | **62.63** | **63.48** | **61.59** |

*Table 4.* Compatibility of OAK with different adaptive optimizers. See Section 6.2 Compatibility of OAK.

| Method | $n = 8$ | $n = 16$ | $n = 32$ | $n = 64$ |
|---|---|---|---|---|
| OAK (AdaGrad) | 31.77 | 33.75 | 34.38 | 37.34 |
| OAK (Adam) | **33.85** | **36.96** | 36.07 | 37.66 |
| OAK (AdamW) | 33.46 | 35.48 | **37.43** | **37.89** |

*Table 5.* Ablation study of Preconditioner Estimation. See Section 6.2 Ablation Study.

| Method | $n = 8$ | $n = 16$ | $n = 32$ | $n = 64$ |
|---|---|---|---|---|
| OAK (SGDM) | 33.46 | 34.65 | 34.98 | 35.85 |
| OAK (AdaGrad) | 31.77 | 33.75 | 34.38 | 37.34 |
| OAK (Adam) | **33.85** | **36.96** | **36.07** | **37.66** |

cause the kernel regime to fail. To validate this, we synthetically control the task discrepancy on the MNLI dataset as follows: for each data point (input text) in finetuning set, we randomly shuffle a fixed percentage of the words. This procedure creates unnatural language (discrepancy tasks) that never existed in pretraining as the percentage increases. We present accuracy with different ratio of shuffled words in Figure 3. The results show that as the task discrepancy increases, the performance gap between finetuning and our OAK widens, which is aligned with our theoretical results.

## 7. Conclusion

In this paper, we proposed OAK by adding the preconditioner of adaptive optimizers into NTK. We also explain when and how the kernel regime fails in foundation model finetuning. Our extensive experiments validate our findings and show the effectiveness of OAK.

## Acknowledgements

This work is supported by National Science Foundation under Award No. IIS-2117902. The views and conclusions are those of the authors and should not be interpreted as representing the official policies of the funding agencies or the government.

## Impact Statement

This paper presents advancements in understanding the training dynamics of foundation models. By providing a more accurate theoretical framework for adaptive optimization, our work contributes to the interpretability of large-scale AI systems. This work primarily focuses on optimization theory and does not directly raise specific ethical concerns or negative societal consequences.

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

# Appendix

# A. Additional Experiments

## A.1. Additional Main Results

Due to page limit, we put the experimental results of Qwen2.5-0.5B model on single-sentense tasks below. See discussions in Section 6.2 in the main paper.

*Table 6.* Accuracy with Qwen2.5-0.5B backbone on Single-Sentence Tasks. For NTK methods, we highlight the best and second-best score with **bold** and underlining. See Section 6.2 Performance Evaluation.

| Method | SST2 | | | | COLA | | | | Avg. Acc. |
|---|---|---|---|---|---|---|---|---|---|
| | $n = 8$ | $n = 16$ | $n = 32$ | $n = 64$ | $n = 8$ | $n = 16$ | $n = 32$ | $n = 64$ | |
| Finetune | 53.39 | 54.02 | 55.66 | 72.98 | 50.95 | 51.92 | 51.01 | 52.54 | 55.31 |
| NTK | 53.19 | **54.30** | 56.71 | 57.55 | 49.15 | **51.17** | 54.04 | **56.90** | 54.13 |
| SignGD-NTK | 52.08 | 52.21 | 55.99 | 55.14 | 47.66 | 49.28 | 52.73 | 54.04 | 52.39 |
| OAK (Adam) | **53.87** | 54.02 | **56.84** | **57.69** | **49.80** | 51.01 | **54.32** | 56.62 | **54.27** |

## A.2. Computational Overhead

Constructing a standard NTK matrix requires per-sample Jacobian computation ($\mathcal{O}(Nd)$) and pairwise inner products ($\mathcal{O}(N^2d)$), giving $\mathcal{O}(N^2d)$ overall. OAK adds only $\mathcal{O}(Nd)$ for preconditioning the gradient features, so it shares the same asymptotic complexity. Tables 7 and 8 report empirical wall-clock time and peak GPU memory for kernel construction on MNLI ($n = 16$) across model scales, confirming OAK adds at most $\sim 10\%$ overhead even at 1.5B parameters.

*Table 7.* Wall-clock time (s) for kernel construction (MNLI, $n = 16$).

| Method | RoBERTa | Qwen2.5-0.5B | Qwen2.5-1.5B |
|---|---|---|---|
| NTK | 23.5 | 64.5 | 125.4 |
| SignGD-NTK | 24.1 | 65.8 | 128.6 |
| OAK (Adam) | 25.3 | 67.5 | 134.3 |

*Table 8.* Peak GPU memory (GB) for kernel construction (MNLI, $n = 16$).

| Method | RoBERTa | Qwen2.5-0.5B | Qwen2.5-1.5B |
|---|---|---|---|
| NTK | 5.38 | 9.80 | 24.36 |
| SignGD-NTK | 5.37 | 9.80 | 24.36 |
| OAK (Adam) | 5.89 | 10.63 | 26.54 |

## A.3. Results on Generative Tasks

Applying NTK to full autoregressive generation remains an open problem, since the $\mathcal{O}(N^2d)$ kernel must be recomputed for each generated token; prior NTK work on language models therefore restricts evaluation to classification. To probe the generative setting, we evaluate single-step next-token prediction, which reduces to classification over the vocabulary, on the LAMA cloze benchmark with Qwen2.5-0.5B ($n = 64$). OAK outperforms the other kernels and nearly matches full finetuning, indicating the framework extends beyond classification.

*Table 9.* Top-1 next-token accuracy (%) on LAMA (Qwen2.5-0.5B, $n = 64$).

| Method | Acc. |
|---|---|
| NTK | 31.4 |
| SignGD-NTK | 33.1 |
| OAK (Adam) | **37.2** |
| Full Adam FT | 37.5 |

## A.4. Downstream Utility: Data Selection

To demonstrate practical value beyond a theoretical tool, we apply OAK to data selection. For MNLI, MRPC, and QQP ($n = 64$), we compute per-sample importance scores using standard NTK, SignGD-NTK, and OAK, select the top $50\%$ ($n = 32$) to form subsets $D_{\text{NTK}}, D_{\text{Sign}}, D_{\text{OAK}}$, and finetune RoBERTa on each. Finetuning on $D_{\text{OAK}}$ achieves performance comparable to the full dataset, confirming OAK's utility for selecting informative samples.

*Table 10.* Accuracy (%) of RoBERTa finetuned on kernel-selected subsets.

| Subset | MNLI | MRPC | QQP |
|---|---|---|---|
| Full FT | 35.5 | 73.2 | 61.6 |
| $D_{\text{NTK}}$ | 34.6 | 71.3 | 61.4 |
| $D_{\text{Sign}}$ | 34.9 | 71.9 | 61.4 |
| $D_{\text{OAK}}$ | **35.3** | 72.8 | 61.5 |

## B. Proofs

### B.1. Proof of Lemma 4.8

Let $\theta_{FT}^*$ be one of the finetuning loss minimizers. By definition, $\nabla \mathcal{L}_{FT}(\theta_{FT}^*) = 0$. Using Assumption 4.3, we can bound the gradient norm at $\theta_0$ as:

$$||\nabla \mathcal{L}_{FT}(\theta_0)||_2 = ||\nabla \mathcal{L}_{FT}(\theta_0) - \nabla \mathcal{L}_{FT}(\theta_{FT}^*)||_2 \leq L_\theta ||\theta_0 - \theta_{FT}^*||_2. \tag{10}$$

Next, we use the triangle inequality to decompose the parameter distance:

$$||\theta_0 - \theta_{FT}^*||_2 \leq ||\theta_0 - \theta_{PT}^*||_2 + ||\theta_{PT}^* - \theta_{FT}^*||_2. \tag{11}$$

The first term on the RHS is the pretraining optimization error, bounded by $\delta_{PT}$. For the second term, we apply the PL condition (Assumption 4.5) at the point $\theta_{PT}^*$:

$$||\theta_{PT}^* - \theta_{FT}^*||_2 \leq \frac{1}{\mu} ||\nabla \mathcal{L}_{FT}(\theta_{PT}^*)||_2. \tag{12}$$

Since $\theta_{PT}^*$ is a minimizer of the pretraining loss, we have $\nabla \mathcal{L}_{PT}(\theta_{PT}^*) = \mathbb{E}_{\xi \sim D_1}[\nabla \mathcal{L}(\xi; \theta_{PT}^*)] = 0$. We can then express the finetuning gradient at $\theta_{PT}^*$ as:

$$||\nabla \mathcal{L}_{FT}(\theta_{PT}^*)||_2 = ||\nabla \mathcal{L}_{FT}(\theta_{PT}^*) - \nabla \mathcal{L}_{PT}(\theta_{PT}^*)||_2 \tag{13}$$

$$= ||\mathbb{E}_{\xi \sim D_2}[\nabla \mathcal{L}(\xi; \theta_{PT}^*)] - \mathbb{E}_{\xi \sim D_1}[\nabla \mathcal{L}(\xi; \theta_{PT}^*)]||_2. \tag{14}$$

Under Assumption 4.4, the gradient $\nabla \mathcal{L}(\xi; \theta_{PT}^*)$ is $L_\xi$-Lipschitz with respect to $\xi$. By the Kantorovich-Rubinstein duality theorem, this difference is bounded by the Wasserstein-1 distance:

$$||\nabla \mathcal{L}_{FT}(\theta_{PT}^*)||_2 \leq L_\xi \cdot W_1(D_1, D_2). \tag{15}$$

Substituting all bounds back into the main inequality yields:

$$||\nabla \mathcal{L}_{FT}(\theta_0)||_2 \leq L_\theta \left( ||\theta_0 - \theta_{PT}^*||_2 + ||\theta_{PT}^* - \theta_{FT}^*||_2 \right) \tag{16}$$

$$\leq L_\theta \left( \delta_{PT} + \frac{1}{\mu} ||\nabla \mathcal{L}_{FT}(\theta_{PT}^*)||_2 \right) \tag{17}$$

$$\leq L_\theta \left( \delta_{PT} + \frac{L_\xi}{\mu} W_1(D_1, D_2) \right). \quad \square \tag{18}$$

### B.2. Proof of Lemma 4.9

Using the triangle inequality, we decompose the norm of the Hessian by introducing $\theta_{PT}^*$:

$$||H_f(\theta_0)||_2 \leq ||H_f(\theta_0) - H_f(\theta_{PT}^*)||_2 + ||H_f(\theta_{PT}^*)||_2. \tag{19}$$

The first term is bounded by invoking the $L_H$-Lipschitz continuity of the Hessian (Assumption 4.6):

$$||H_f(\theta_0) - H_f(\theta_{PT}^*)||_2 \leq L_H ||\theta_0 - \theta_{PT}^*||_2 \leq L_H \cdot \delta_{PT}. \tag{20}$$

The second term is directly bounded by Assumption 4.7, so $||H_f(\theta_{PT}^*)||_2 \leq C_{PT}$. Combining these two bounds gives the result:

$$||H_f(\theta_0)||_2 \leq L_H \cdot \delta_{PT} + C_{PT}. \quad \square \tag{21}$$

### B.3. Proof of Theorem 4.10

The linearization error is captured by the remainder term of the second-order Taylor expansion of $f(\xi; \theta_T)$ around $\theta_0$. By Taylor's theorem, this remainder $R$ is bounded as:

$$||R||_2 \leq \frac{1}{2} ||\theta_T - \theta_0||_2^2 \sup_{\theta \in [\theta_0, \theta_T]} ||H_f(\theta)||_2. \tag{22}$$

To bound this error, we first bound the parameter drift $||\theta_T - \theta_0||_2$ and the Hessian norm.

Let $D_t = \max_{0 \le i \le t} ||\theta_i - \theta_0||_2$ be the maximum parameter drift from the initial point $\theta_0$ up to step $t$. For any step $i \le T$, the gradient norm is bounded by applying Assumption 4.3:

$$||\nabla \mathcal{L}_{FT}(\theta_{i-1})||_2 \le ||\nabla \mathcal{L}_{FT}(\theta_0)||_2 + L_\theta ||\theta_{i-1} - \theta_0||_2 \le \epsilon_g + L_\theta D_{i-1}. \tag{23}$$

The total drift after $T$ steps of gradient descent ($\theta_i = \theta_{i-1} - \eta \nabla \mathcal{L}_{FT}(\theta_{i-1})$) is bounded by the sum of single-step updates:

$$||\theta_T - \theta_0||_2 = \left\| \sum_{i=1}^{T} (\theta_i - \theta_{i-1}) \right\|_2 \le \sum_{i=1}^{T} \eta ||\nabla \mathcal{L}_{FT}(\theta_{i-1})||_2 \le \sum_{i=1}^{T} \eta(\epsilon_g + L_\theta D_{i-1}). \tag{24}$$

Since $D_{i-1} \le D_T$ for all $i \le T$, we can establish a self-consistent inequality for the maximum drift $D_T$:

$$D_T \le T\eta(\epsilon_g + L_\theta D_T) = T\eta\epsilon_g + T\eta L_\theta D_T. \tag{25}$$

Rearranging the inequation, we solve $D_T$, which is valid for $1 - T\eta L_\theta > 0$, as:

$$D_T \le \frac{T\eta\epsilon_g}{1 - T\eta L_\theta}. \tag{26}$$

Next, we bound the Hessian norm over the path. Using Assumption 4.6 (Lipschitz Hessian) and the triangle inequality:

$$\sup_{\theta \in [\theta_0, \theta_T]} ||H_f(\theta)||_2 \le ||H_f(\theta_0)||_2 + L_H \sup_{\theta \in [\theta_0, \theta_T]} ||\theta - \theta_0||_2 \le \epsilon_f + L_H D_T. \tag{27}$$

Finally, we substitute the bounds for the drift ($||\theta_T - \theta_0||_2 \le D_T$) and the Hessian back into the remainder inequality:

$$||R||_2 \le \frac{1}{2} D_T^2 (\epsilon_f + L_H D_T) \tag{28}$$

$$\le \frac{1}{2} \left( \frac{T\eta\epsilon_g}{1 - T\eta L_\theta} \right)^2 \left( \epsilon_f + L_H \frac{T\eta\epsilon_g}{1 - T\eta L_\theta} \right). \quad \square \tag{29}$$

## B.4. Proof of Theorem 4.11

The change in features from initialization to step $T$ can be bounded by:

$$||\nabla_\theta f(\xi; \theta_T) - \nabla_\theta f(\xi; \theta_0)||_2 \le \sup_{\theta \in [\theta_0, \theta_T]} ||H_f(\theta)||_2 \cdot ||\theta_T - \theta_0||_2. \tag{30}$$

Let $D_T$ be the maximum parameter drift over $T$ steps, as derived in Eq. (26) from the proof of Theorem 4.10. We can bound the two terms on the right-hand side of the inequality. The parameter drift is bounded by $D_T$:

$$||\theta_T - \theta_0||_2 \le D_T \le \frac{T\eta\epsilon_g}{1 - T\eta L_\theta}. \tag{31}$$

The Hessian norm is bounded using its Lipschitz continuity (Assumption 4.6) and the definition of $D_T$:

$$\sup_{\theta \in [\theta_0, \theta_T]} ||H_f(\theta)||_2 \le ||H_f(\theta_0)||_2 + L_H \sup_{\theta \in [\theta_0, \theta_T]} ||\theta - \theta_0||_2 \le \epsilon_f + L_H D_T. \tag{32}$$

Substituting these bounds back into the main inequality gives the total deviation of the features:

$$||\nabla_\theta f(\xi; \theta_T) - \nabla_\theta f(\xi; \theta_0)||_2 \le (\epsilon_f + L_H D_T) \cdot D_T. \tag{33}$$

Finally, substituting the expression for $D_T$ yields the explicit bound:

$$||\nabla_\theta f(\xi; \theta_T) - \nabla_\theta f(\xi; \theta_0)||_2 \le \left( \epsilon_f + L_H \frac{T\eta\epsilon_g}{1 - T\eta L_\theta} \right) \left( \frac{T\eta\epsilon_g}{1 - T\eta L_\theta} \right). \quad \square \tag{34}$$

### B.5. Proof of Theorem 5.2

We use the triangle inequality to decompose the total error into two components: a sampling error and a dynamic error.

$$||\hat{m}_0 - m_T||_2 \leq ||\hat{m}_0 - \nabla \mathcal{L}_{FT}(\theta_0)||_2 + ||\nabla \mathcal{L}_{FT}(\theta_0) - m_T||_2. \tag{35}$$

**Bounding the Sampling Error:** The first term in Eq. (35) measures the error from using a finite sample set of size $N$. The estimator $\hat{m}_0$ is the sample mean of $N$ i.i.d. random vectors, with $\mathbb{E}[\hat{m}_0] = \nabla \mathcal{L}_{FT}(\theta_0)$. Under Assumption 5.1, the variance of the estimator is bounded: $\mathbb{E}[||\hat{m}_0 - \nabla \mathcal{L}_{FT}(\theta_0)||_2^2] \leq \frac{\sigma^2}{N}$. By applying Chebyshev's inequality, we have that with probability at least $(1 - \delta)$:

$$||\hat{m}_0 - \nabla \mathcal{L}_{FT}(\theta_0)||_2 \leq \frac{\sigma}{\sqrt{N\delta}}. \tag{36}$$

**Bounding the Dynamic Error:** The second term in Eq. (35) captures the divergence between the true initial gradient and the true momentum after $T$ steps. For simplification, let $g_t = \nabla \mathcal{L}_{FT}(\theta_{t-1})$. The true Adam momentum after $T$ steps, with $m_0 = 0$, is $m_T = (1 - \beta_1) \sum_{i=1}^{T} \beta_1^{T-i} g_i$. We can express the initial gradient $g_1 = \nabla \mathcal{L}_{FT}(\theta_0)$ using the identity $1 = (1 - \beta_1^T) + \beta_1^T = (1 - \beta_1) \sum_{i=1}^{T} \beta_1^{T-i} + \beta_1^T$. This allows us to write the difference as:

$$g_1 - m_T = \left( (1 - \beta_1) \sum_{i=1}^{T} \beta_1^{T-i} g_1 + \beta_1^T g_1 \right) - (1 - \beta_1) \sum_{i=1}^{T} \beta_1^{T-i} g_i \tag{37}$$

$$= (1 - \beta_1) \sum_{i=1}^{T} \beta_1^{T-i} (g_1 - g_i) + \beta_1^T g_1. \tag{38}$$

Taking the norm and applying the triangle inequality yields:

$$||g_1 - m_T||_2 \leq (1 - \beta_1) \sum_{i=1}^{T} \beta_1^{T-i} ||g_1 - g_i||_2 + \beta_1^T ||g_1||_2. \tag{39}$$

From Lemma 4.8, we have $||g_1||_2 \leq \epsilon_g$. From Assumption 4.3, $||g_1 - g_i||_2 \leq L_\theta ||\theta_0 - \theta_{i-1}||_2$. Since $||\theta_0 - \theta_{i-1}||_2$ is bounded by the maximum drift $D_T$ (from Eq. (26)) for any $i \leq T$, we have:

$$||g_1 - m_T||_2 \leq (1 - \beta_1) \left( \sum_{i=1}^{T} \beta_1^{T-i} \right) (L_\theta D_T) + \beta_1^T \epsilon_g \tag{40}$$

$$= (1 - \beta_1^T) L_\theta D_T + \beta_1^T \epsilon_g. \tag{41}$$

Combining the sampling and dynamic error bounds completes the proof. $\square$

### B.6. Proof of Theorem 5.4

We again decompose the error into sampling and dynamic parts:

$$||\hat{v}_0 - v_T||_2 \leq ||\hat{v}_0 - \mathbb{E}[g_1^2]||_2 + ||\mathbb{E}[g_1^2] - v_T||_2, \tag{42}$$

where $g_1^2$ denotes the element-wise square of a per-sample gradient $\nabla \mathcal{L}(\xi; \theta_0)$, and $\mathbb{E}[g_1^2]$ is its true expectation at initialization.

**Bounding the Sampling Error:** The estimator $\hat{v}_0$ is the sample mean of $N$ random vectors $g_i^2$. The variance of this estimator is $\mathrm{Var}(\hat{v}_0) = \frac{1}{N} \mathrm{Var}(g_1^2)$. By definition, $\mathrm{Var}(||g_1^2||_2) = \mathbb{E}[||g_1^2||_2^2] - (\mathbb{E}[||g_1^2||_2])^2 = \mathbb{E}[||\nabla \mathcal{L}(\xi; \theta_0)||_2^4] - ||\mathbb{E}[g_1^2]||_2^2 \leq M - (\sigma^2 + \epsilon_g^2)^2$. Applying Chebyshev's inequality gives the sampling error bound with probability at least $(1 - \delta)$.

**Bounding the Dynamic Error:** We analyze the term $||\mathbb{E}[g_1^2] - v_T||_2$. The true variance is $v_T = (1 - \beta_2) \sum_{i=1}^{T} \beta_2^{T-i} g_i^2$, where $g_i = \nabla \mathcal{L}_{FT}(\theta_{i-1})$. Following the same logic as in Theorem 5.2, we have:

$$||\mathbb{E}[g_1^2] - v_T||_2 \leq (1 - \beta_2^T) \sup_{i \leq T} ||\mathbb{E}[g_1^2] - g_i^2||_2 + \beta_2^T ||\mathbb{E}[g_1^2]||_2. \tag{43}$$

The term $||\mathbb{E}[g_1^2]||_2$ is bounded by $\mathbb{E}[||g_1||_2^2] = \text{Var}(g_1) + (\mathbb{E}[g_1])^2 \leq \sigma^2 + \epsilon_g^2$. For the core term, we bound the difference between squared gradients:

$$||g_1^2 - g_i^2||_2 = ||(g_1 - g_i) \odot (g_1 + g_i)||_2 \leq ||g_1 - g_i||_2 ||g_1 + g_i||_2 \tag{44}$$

$$\leq ||g_1 - g_i||_2 (||g_1||_2 + ||g_i||_2) \leq ||g_1 - g_i||_2 (2||g_1||_2 + ||g_i - g_1||_2). \tag{45}$$

Using Lemma 4.8 ($||g_1||_2 \leq \epsilon_g$) and Assumption 4.3 ($||g_1 - g_i||_2 \leq L_\theta D_T$), we get:

$$||g_1^2 - g_i^2||_2 \leq (L_\theta D_T)(2\epsilon_g + L_\theta D_T) = 2\epsilon_g L_\theta D_T + (L_\theta D_T)^2. \tag{46}$$

This bound on the difference of full-batch gradients also holds for the difference involving the expectation $\mathbb{E}[g_1^2]$. Substituting these pieces back into the dynamic error inequality yields:

$$||\mathbb{E}[g_1^2] - v_T||_2 \leq (1 - \beta_2^T)\left(2\epsilon_g L_\theta D_T + (L_\theta D_T)^2\right) + \beta_2^T(\sigma^2 + \epsilon_g^2). \tag{47}$$

Combining the two error bounds completes the proof. $\qquad\square$

