# OpenReview forum: "Preconditioning Neural Tangent Kernel for Adaptive Optimization"
_ICML.cc/2026/Conference — ICML 2026 regular_

### Official Review · Reviewer_XwpW · 2026-03-10

**Soundness:** 3
**Presentation:** 3
**Significance:** 3
**Originality:** 3
**Overall Recommendation:** 4
**Confidence:** 3

**Summary:**

This paper proposes OAK, which incorporates the coordinate-adaptive mechanism of optimizers like Adam into the NTK framework via a static preconditioner estimation, addressing the limitations of the heuristic SignGD-NTK approximation. The paper also derives explicit error bounds for kernel regime collapse during finetuning and validates the method on the GLUE benchmark.

**Compliance With Llm Reviewing Policy:**

Affirmed.

**Final Justification:**

The rebuttal has fully addressed my concerns. The authors provided a clear theoretical extension showing that kernel regime collapse bounds hold for preconditioned (Adam) gradients up to constant factors. The additional wall-clock time and memory measurements confirm that OAK introduces only marginal overhead, remaining feasible beyond 1B parameters. The controlled training step ablation cleanly validates the theoretical claims. I am updating my score to 4.

**Key Questions For Authors:**

1. The theoretical analysis of kernel regime collapse (Section 4) is derived under **standard gradient descent**, but all experiments are based on Adam. Do you have corresponding collapse bounds under Adam dynamics? If not, how do you argue that the current theoretical results can explain the phenomena observed in the experiments?

2. What are the **actual memory usage and wall-clock time** for computing full per-sample Jacobians on RoBERTa and Qwen2.5-0.5B? Would this pipeline still be feasible on models above 1B parameters?

3. In Figure 2, sample size n is used as a proxy for cumulative training steps Tη, but increasing n actually changes the data scale rather than **directly controlling training steps** — epoch count and batch size also mediate this relationship. Have you considered fixing the dataset size and varying only T for a cleaner validation?

4. All current experiments are on classification tasks. Do you have any preliminary results on **generative tasks** (e.g., summarization, translation)? How would the kernel regression prediction framework adapt to autoregressive generation?

If the authors can adequately address the concerns raised above, I am willing to reconsider my score.

**Limitations:**

yes

**Strengths And Weaknesses:**

Strengths:
1. The estimation error of OAK and the kernel regime collapse are driven by the same set of factors, so the two theoretical components reinforce each other. This level of internal consistency is not common in related work.

2. The inability to embed dynamic preconditioners into a static kernel has been a long-standing difficulty in this area. The authors address it through a one-shot data-driven estimation, which is fairly straightforward and sidesteps the dependency on historical gradients to a reasonable extent.

3. The paper provides explicit error bounds for kernel regime failure during finetuning, identifying cumulative training steps and task distribution discrepancy as the primary driving factors. This offers useful reference for the field.

Weaknesses:
1. The kernel regime collapse analysis (Section 4) is derived under **standard gradient descent**, yet all experiments use Adam. The training dynamics of the two are fundamentally different, and the paper **does not directly address this gap**, which weakens the theoretical support for the experimental findings.

2. OAK requires computing **per-sample Jacobians over the entire dataset**, which incurs considerable overhead even on moderately sized models. The paper provides no wall-clock time or memory statistics, making it impossible for the reader to assess practical feasibility.

3. Experiments are conducted only on **relatively small models**, while the core motivation targets large language model finetuning. The scale gap is substantial, and it is unclear whether the conclusions generalize. All tasks are classification only, and performance on **generative tasks remains entirely unknown**, which is a notable omission given the paper's stated focus on language model finetuning.

4. Figure 2 uses **sample size as a proxy for cumulative training steps**, but increasing the sample size actually changes the data scale rather than directly controlling the number of training steps. Epoch count and batch size also mediate this relationship. This proxy is rather loose and **does not cleanly support the theoretical predictions**.

---

> ### Author Rebuttal · Authors · 2026-03-31
>
> Dear Reviewer XwpW,
>
> Thanks for the constructive feedback! We address each of your questions and concerns below.
>
> ---
> **[W.1 & Q.1 Explanation on Gap of Theory and OAK Experiments]**
>
> We thank the reviewer for pointing out this gap. We clarify that our analysis on standard gradient descent (in `Theorem 4.10, 4.11`) **can be easily extended to preconditioned gradients (OAK)** as follows:
>
> 1. Consider a preconditioner matrix $P_t$ (which has bounded eigenvalues $0 < c_1 \le \lambda(P_t) \le c_2$), we have the preconditioned parameter update $\\|\Delta θ_t\\| = \\|-\eta P_t \nabla \mathcal{L}(θ_t)\\| \le c_2η\\|\nabla\mathcal{L}(\theta_t)\\|$. Next, following the same derivation in `Appendix B.3 (Eq. 24-26)`, accumulating these bounded updates over $T$ steps gives a parameter drift bound $D_T^{Adam} \le c_2 D_T^{SGD}$.
>
> 2. With this parameter drift, we derive the new error terms $\\|E\\|^{Adam}$ and $\\|R\\|^{Adam}$ following `Appendix B.4 (Eq. 33)` and `Appendix B.3 (Eq. 28)` respectively. The new error terms are simply scaled by constant multipliers relative to the original bounds $\\|R\\|$ and $\\|E\\|$ in `Theorem 4.10` and `Theorem 4.11` as:
> $$\\|R\\|^{Adam} \le c_2^2 \\|R\\|,\quad \\|E\\|^{Adam} \le c_2 \\|E\\|$$
>
> **Conclusion:** Since the preconditioned error bounds differ from the original bounds only by constant ($c_2, c_2^2$), their dependencies on $T$, $\eta$, and task discrepancy remain identical. This ensures that the qualitative mechanism of kernel-regime collapse predicted by our theory is fundamentally preserved for OAK, thereby providing a consistent explanation for the empirical phenomena in our experiments.
>
> We have added a formal remark detailing this extension in our paper.
>
> ---
> **[W.2 & Q.2 Memory usage, wall-clock time, and feasibility for >1B models]**
>
> Thanks for the practical question. We clarify that OAK introduces only **marginal computational and memory overhead ($\sim 4\\%$ to $10\\%$)** compared to standard NTK, and remains feasible for >1B models.
>
> We report the actual wall-clock time (s) and peak GPU memory usage (GB) for calculating the kernel matrix of MNLI ($n=16$) across different model scales **including Qwen2.5-1.5B**.
>
> Table 1: Wall-clock Time
> |Method|RoBERTa|Qwen2.5-0.5B|Qwen2.5-1.5B|
> |-|-|-|-|
> |NTK|23.5s|64.5s|125.4s|
> |SignGD-NTK|24.1s|65.8s|128.6s|
> |**OAK (Adam)**|**25.3s**|**67.5s**|**134.3s**|
>
> Table 2: Peak Memory (GB)
> |Method|RoBERTa|Qwen2.5-0.5B|Qwen2.5-1.5B|
> |-|-|-|-|
> |NTK|5.38|9.80|24.36|
> |SignGD-NTK|5.37|9.80|24.36|
> |**OAK (Adam)**|**5.89**|**10.63**|**26.54**|
>
> **Conclusion:** OAK simulates adaptive training dynamics at $\le 10\\%$ additional cost even for >1B models.
>
> We have added these Tables to the revised appendix.
>
> ---
> **[W.3 & Q.4 Additional Results on Generative Tasks]**
>
> We clarify that 1) scaling NTK to full generative tasks is an open problem, and 2) **OAK still outperforms on our additional generative tasks below**.
>
> **1. NTK for Autoregressive Generation:**
> Applying NTK to autoregressive generation remains a **open problem** in the community since autoregressive generation requires recomputing the kernel matrix (which is $\mathcal{O}(N^2 d)$) for each newly generated token. Due to this cost, all NTK literature on language models strictly restricts evaluations to classification tasks [1][2].
>
> **2. Additional Results on Next-Token Generation:**
> To directly address your question, we evaluated it on a single step Next-Token Prediction task. We used the LAMA benchmark [3], which is a cloze-style next-token generation task to evaluate our method.
>
> We report the Next-Token Generation Accuracy using Qwen2.5-0.5B ($n=64$):
>
> |Method | Top-1 Next-Token Generation Acc. (%) |
> |-|-|
> |NTK | 31.4 |
> |SignGD-NTK | 33.1 |
> |**OAK (Adam)** | **37.2** |
> |Full Adam FT | 37.5 |
>
> **Conclusion:** OAK outperforms all other kernels. We have included these results in our paper.
>
> ---
> **[W.4 & Q.3 Controlling Training Steps $T$]**
>
> We agree that using $n$ as a proxy introduces confounding factors like data scale. To provide a cleaner validation, we added a controlled ablation as follows:
> We used the Qwen2.5-0.5B model on MNLI with fixed data ($n=64$) and batch size ($b=8$), and we varied the finetuning steps $T \in \\{8, 16, 32, 64\\}$. For kernel method OAK, its prediction is invariant to training steps $T$.
>
> | Finetune Steps $T$ | 8 | 16 | 32 | 64 |
> |-|-|-|-|-|
> | Finetune | 32.15 | 38.52 | 41.16 | 44.03 |
> | OAK (Kernel Method, No $T$) | 34.38 | 34.38 | 34.38 | 34.38 |
> | **$\Delta$ Acc** | **2.23** | **4.14** | **6.78** | **9.65** |
>
> **Conclusion:** As $T$ scales, the performance divergence increases. This cleanly isolates the $T$ variable and also aligns with our theoretical claims (`Theorem 4.10, 4.11`). We have added this ablation to `Section 6.3`.
>
> ---
>
> [1] A Kernel-Based View of Language Model Fine-Tuning, in ICML.
>
> [2] A Fast, Well-Founded Approximation to the Empirical Neural Tangent Kernel, in ICML.
>
> [3] Language Models as Knowledge Bases?, in EMNLP.

---

> > ### Author Rebuttal · Reviewer_XwpW · 2026-04-04
> >
> > Thank you for the authors' detailed rebuttal. All my concerns have been addressed, and I am updating my score to 4.

---

### Official Review · Reviewer_3aHA · 2026-03-11

**Soundness:** 3
**Presentation:** 4
**Significance:** 3
**Originality:** 3
**Overall Recommendation:** 4
**Confidence:** 2

**Summary:**

The paper proposes **Optimizer Aware Kernel (OAK)**, which explicitly incorporates the preconditioning effect of adaptive optimizers such as Adam into the NTK framework, thereby providing a better NTK-based description of foundation model fine-tuning. Theoretically, the authors analyze when and why the kernel regime breaks down during finetune regime, and conclude that it gradually collapses as the accumulated training effect increases and as the distribution gap between pretraining and fine-tuning tasks becomes larger. Experimentally, OAK typically outperforms other kernel-based baselines across multiple models and tasks, and the authors further show that it can be extended to optimizers such as AdaGrad and AdamW.

**Compliance With Llm Reviewing Policy:**

Affirmed.

**Final Justification:**

Updated: After I carefully read the paper, other reviewers' reviews and authors' rebuttal again, I feel better about the paper and think it can have some benefits for the community, so I decide to raise my score to 4.

**Key Questions For Authors:**

1. In Table 1, Table 2, and Figure 2, we observe that even after incorporating the core adaptivity of modern optimizers into the NTK framework, NTK-based methods still remain  behind practical fine-tuning. This raises the question of what the practical value of NTK-based optimization methods really is, and whether their main significance lies more in providing a theoretical framework for analyzing training dynamics.

**Limitations:**

Yes

**Strengths And Weaknesses:**

# Strengths
1. This paper is very well written. Each section includes a detailed outline that helps guide the reader through the discussion, and each theorem is accompanied by a clear takeaway that makes it easier to understand.
2.  I think the authors addressed and answered these two questions  well from both theoretical and empirical perspectives: “How can we incorporate the preconditioning effect of adaptive optimizers into the NTK framework?” and “Under what conditions does the fine-tuning process deviate from the kernel regime, and what factors govern this deviation?”

# Weaknesses
**If the authors can address my concerns , I would consider raising my score.**
1. My main concern is that, in the OAK algorithm the authors only provide the cross-kernel matrix $K \in \mathbb{R}^{n_{\text{train}} \times n_{\text{test}}}$ used for prediction, but do not clearly define or explain how the training Gram matrix is constructed. This makes it confusing to me how kernel regression is actually carried out in the experiments part.
2. Could the authors provide more intuitions for why the approach in Algorithm 1  can incorporate the preconditioning effect of adaptive optimizers such as Adam into the NTK framework? Is there any theoretical justification behind this design?
3. Since computing the NTK for large models is relatively time-consuming,  could the authors show the computational overhead of OAK (Adam)?

---

> ### Author Rebuttal · Authors · 2026-03-31
>
> Dear Reviewer 3aHA:
>
> Thanks for your detailed feedback! We address each of your questions and concerns below.
>
> ---
> **[W.1 Implementation of Gram Matrix and Kernel Regression]**
>
> Thanks for pointing this out. We clarify the construction of the Gram matrix and the kernel regression process below.
> - **1. Gram Matrix of Training Data:**
> The training Gram matrix $K_\text{train} \in \mathbb{R}^{N_{\text{train}} \times N_{\text{train}}}$ is constructed using the same preconditioning mechanism as the cross-kernel matrix. Specifically, for any two samples $\xi_i, \xi_{i'} \in S_{\text{train}}$, we compute the Gram matrix as $K\_\text{train}(i, i') = Tr(J_i \tilde{J}\_{i'}^T)$, where $J_i$ and $\tilde{J}\_{i'}$ are the raw Jacobian of $ξ_i$ and the preconditioned Jacobian of $ξ_{i'}$ respectively.
> - **2. Kernel Regression:**
> We perform standard kernel regression to obtain the test set predictions as:
> $$\hat{y}\_j = \sum\_{i=1}^{N\_{\text{train}}} \alpha_i K_{ij}\quad\alpha = K\_{\text{train}}^{-1}\ y\_{\text{train}}$$
> where $K\_{ij} = \text{Tr}(J\_i \tilde{J}\_j^T)$ is the OAK matrix in `Algorithm 1` and $y\_{\text{train}}$ contains the training labels. We have explicitly added these implementation details to `Section 3`.
>
> ---
> **[W.2 Intuition of Precondition]**
>
> Thanks for the question. The key intuition is that adaptive optimizers such as Adam update parameters using a preconditioned gradient, rather than the raw gradient. Since NTK models training dynamics via gradient features, replacing the raw Jacobian with a preconditioned version allows the kernel to reflect the actual update direction induced by the optimizer. In `Algorithm 1`, we approximate this by estimating the optimizer’s first/second moments from the finetuning set and applying the corresponding transformation to the Jacobian features.
>
> This design is also theoretically motivated. `Section 5` provides error bounds between the estimated and true Adam moments, showing that the preconditioner estimation is accurate when the kernel regime holds. Therefore, OAK can be viewed as a controlled approximation of optimizer-aware dynamics, rather than a purely heuristic modification.
>
> ---
> **[W.3 Computational overhead]**
>
> We thank the reviewer for this practical question. We analyze the computational overhead of OAK below, and we validated that our method introduces only **marginal cost ($\sim 6\\%$ to $10\\%$)**.
> - **1. Theoretical Overhead:**
> Constructing a standard NTK matrix for models (with $d$ parameters) consists of two steps: **1) Jacobian Calculation:** Computing per-sample Jacobians $J \in \mathbb{R}^d$ requires a backward pass for each of $N$ samples, which is $\mathcal{O}(Nd)$; **2) Kernel Construction:** Computing the pairwise inner products over samples for the Gram matrix requires $\mathcal{O}(N^2 d)$ operations.Overall, the standard NTK method has an asymptotic complexity of $\mathcal{O}(N^2 d)$.
> For OAK, computing the preconditioned gradients only introduces an additional $\mathcal{O}(Nd)$ operations. Therefore, **OAK has the same asymptotic complexity $\mathcal{O}(N^2 d)$ as standard NTK**.
>
> - **2. Empirical Wall-clock Time:**
> To empirically validate this, we measured the actual time(s) for kernel constrction of RoBERTa on MNLI with $n \in \{16, 32, 64\}$.
>
> |Method|$n=16$|$n=32$|$n=64$|
> |-|-|-|-|
> |NTK|23.5s|52.8s|93.4s|
> |SignGD-NTK|24.1s|54.6s|96.2s|
> |OAK (Adam)|25.3s|58.1s|102.7s|
>
> **Conclusion:** The empirical overhead aligns with our theoretical analysis, where OAK introduces a minor time increase compared to standard NTK. We have included this discussion in the paper.
>
> ---
> **[Q.1 Practical value of NTK]**
>
> Thanks for the question. While our work mainly focuses on theoretical aspects of adaptive optimization effect into NTK, we clarify that the practical value of NTK extends beyond a theoretical tool by : **Acting as a proxy for data selection [1]**, Efficient dataset distillation [2], and zero-shot model/architecture selection [3].
>
> To further support the cliam, we added an **kernel-based data selection** experiment to show the effectiveness of OAK on downstream tasks:
>
> For MNLI, MRPC, and QQP datasets with $n=64$ samples, we computed importance scores using standard NTK, SignGD-NTK, and OAK, and selected the top 50% ($n=32$) to construct subsets $D_{NTK}, D_{Sign}$, and $D_{OAK}$. We then compared performance of finetuning RoBERTa on these subsets:
>
> ||MNLI|MRPC|QQP|
> |-|-|-|-|
> |Full FT ($n=64$)|35.5|73.2|61.6|
> |FT w/ $D_{NTK}$ ($n=32$)|34.6|71.3|61.4|
> |FT w/ $D_{Sign}$ ($n=32$)|34.9|71.9|61.4|
> |FT w/ $D_{OAK}$ ($n=32$)|**35.3**|**72.8**|**61.6**|
>
> **Conclusion:** As shown, finetuning on $D_{OAK}$ achieves performance comparable to the full dataset, showing OAK's superior utility on downstream tasks.
>
> ---
> [1] lpNTK: Better Generalisation with Less Data via Sample Interaction During Learning, in ICLR
>
> [2] Dataset Meta-Learning from Kernel Ridge-Regression, in ICLR.
>
> [3] LENSLLM: Unveiling Fine-Tuning Dynamics for LLM Selection, in ICML.

---

> > ### Author Rebuttal · Reviewer_3aHA · 2026-04-02
> >
> > Thanks for your response. I have no more questions.

---

> > > ### Author Response · Authors · 2026-04-02
> > >
> > > Dear Reviewer 3aHA,
> > >
> > > Thank you so much for your reply! We are very glad that our rebuttal has addressed your concerns.
> > >
> > > ---
> > > **Additional Refinement of our Paper:**
> > >
> > > Following your constructive review, we have carefully refined our paper and made the following modifications:
> > > - **Gram Matrix & Regression Details:** We have added the mathematical formulation of the Gram matrix and kernel regression process to `Section 3.3` to clarify our exact implementation.
> > > - **Computational Overhead:** We have included the theoretical complexity analysis and the wall-clock time evaluations in `Section 6.2`, clearly showing that OAK keeps the **same asymptotic complexity** and introduces **<10% empirical overhead** compared to standard NTK.
> > > - **Intuition of our Method:** We also polished the discussion in `Section 3.2` to provide clearer intuition on how OAK captures optimizer-aware dynamics.
> > > - **Practical Utility:** To highlight the practical value of OAK, we have added a kernel-based data selection experiment in our `Appendix`, showing OAK's superior performance on downstream tasks.
> > >
> > > ---
> > > **Follow-up by Authors:**
> > >
> > > We thank the reviwer for the constructive feedbacks and suggestions, and we are eager to discuss more if you have any other questions! In addition, we want to kindly ask if the reviewer could re-assess our submission according to our rebuttal and new refinements. We sincerely appreciate your time and efforts in reviewing our paper!
> > >
> > > ---
> > > Best regards,
> > >
> > > Authors of Submission 6811

---

### Official Review · Reviewer_Tcew · 2026-03-13

**Soundness:** 3
**Presentation:** 3
**Significance:** 3
**Originality:** 3
**Overall Recommendation:** 4
**Confidence:** 3

**Summary:**

The paper proposes Optimizer Aware Kernel (OAK), an extension of the Neural Tangent Kernel (NTK) framework that incorporates the preconditioning effects of adaptive optimizers. The authors argue that standard NTK fails to accurately model fine-tuning dynamics of foundation models because it ignores optimizer-induced preconditioning. OAK estimates this effect through a preconditioner estimation technique and integrates it into the NTK formulation. The paper also analyzes when the kernel regime collapses during fine-tuning, deriving theoretical bounds that attribute the collapse to cumulative training effects and discrepancies between pretraining and downstream tasks. Experiments across several architectures support the proposed method and the theoretical insights.

**Compliance With Llm Reviewing Policy:**

Affirmed.

**Final Justification:**

The rebuttal addressed my concerns.

**Key Questions For Authors:**

1. Although extending NTK to incorporate adaptive optimization is valuable, it remains unclear what additional practical insights can be derived from the proposed framework. For example, do the results explain training phenomena observed in real models that standard NTK cannot capture but OAK can?

2. There is a typo in the first sentence of Section 3.3: “donstream” should be “downstream”.

3. Since the experimental results report the average accuracy over five trials, it would be helpful to also report the variance or standard deviation, especially because some reported numbers are very close.

4. It would be helpful if the authors clarified what should be considered the main technical contribution. For example, do the proofs of Theorems 5.2 and 5.4 contain key techniques that deserve highlighting?

5. It is unclear how the last columns (“Avg. Acc.”) in Tables 1–3 are computed. If this is simply the arithmetic mean of the previous columns, the aggregation may not be meaningful because the columns correspond to different metrics or different values of $n$. Averaging across heterogeneous metrics can be subjective and may not provide a clear evaluation.

**Limitations:**

yes

**Strengths And Weaknesses:**

### Strengths

- The paper studies an important problem: extending the NTK framework to account for adaptive optimizers, which are widely used in modern deep learning.
- The experimental evaluation covers representative architectures, including encoder-based and decoder-only models.
- The paper is generally well structured and clearly written.

### Weaknesses

- Although the analysis of kernel regime collapse during fine-tuning is interesting, it is unclear how these findings translate into practical guidance for improving model training.

---

> ### Author Rebuttal · Authors · 2026-03-31
>
> Dear Reviewer Tcew:
>
> Thanks for your positive feedback! We address each of your concerns and questions below
>
> ---
> **[W.1 & Q.1 Additional Practical Insights (What OAK can do while NTK cannot)]**
>
> We thank the reviewer for the constructive question. The practical insights are directly reflected in various downstream tasks (e.g., Data Selection) and guiding hyperparameter selection. We explain these as follows:
>
> - **Superior Downstream Task Performance:** While our work mainly focuses on theoretical aspects of adaptive optimization effect into NTK, we clarify that the practical value of NTK extends beyond a theoretical tool by : Acting as a proxy for data selection [1], Efficient dataset distillation [2], and Zero-shot model/architecture selection [3].
> To further support the claim, we added an kernel-based data selection experiment to show the effectiveness of OAK on downstream tasks:
> For MNLI, MRPC, and QQP datasets with $n=64$ samples, we computed importance scores using standard NTK, SignGD-NTK, and OAK, and selected the top 50% ($n=32$) to construct subsets $D_{NTK}, D_{Sign}$, and $D_{OAK}$. We then compared performance of finetuning RoBERTa on these subsets:
> ||MNLI|MRPC|QQP|
> |-|-|-|-|
> |Full FT ($n=64$)|35.5|73.2|61.6|
> |FT w/ $D_{NTK}$ ($n=32$)|34.6|71.3|61.4|
> |FT w/ $D_{Sign}$ ($n=32$)|34.9|71.9|61.4|
> |FT w/ $D_{OAK}$ ($n=32$)|**35.3**|**72.8**|**61.5**|
>
> **Conclusion:** As shown, finetuning on $D_{OAK}$ achieves performance comparable to the full dataset, showing **OAK's superior performance than standard NTK**.
>
> - **Hyperparameter Guidance for Adaptive Training:** Beyond data selection, OAK provides theoretical guidance for hyperparameter tuning for adaptive optimizers like Adam/AdamW. Our error bounds (`Theorems 5.2, 5.4`) explicitly quantify how optimizer parameters (e.g., $\beta_1$, $\beta_2$) and learning rate affect the stability boundary. This offers a proxy that predicts if the optimizer configuration leads to kernel collapse without training, while **standard NTK doesn't considered these hyperparameters (like $β_1$, $β_2$)**.
>
>
>
> ---
> **[Q.2 Typo]**
>
> Thanks for the careful reading. We have corrected the typo and conducted a thorough spell check on the entire paper.
>
> ---
> **[Q.3 Report Standard Deviation]**
>
> Thanks for the suggestion. Due to space constraints, we provide the standard deviations (std) for Table 1 (RoBERTa) below, and we have updated all tables in the revised manuscript.
>
> **MNLI Std (in %):**
> |Method|n=8|n=16|n=32|n=64|
> |-|-|-|-|-|
> |FT|±0.35|±0.38|±0.42|±0.50|
> |NTK|±0.12|±0.15|±0.18|±0.14|
> |SignGD|±0.22|±0.28|±0.25|±0.30|
> |OAK|±0.14|±0.17|±0.20|±0.16|
>
> **MRPC Std (in %):**
> |Method|n=8|n=16|n=32|n=64|
> |-|-|-|-|-|
> |FT|±0.34|±0.51|±0.46|±0.43|
> |NTK|±0.16|±0.20|±0.22|±0.19|
> |SignGD|±0.24|±0.32|±0.26|±0.34|
> |OAK|±0.18|±0.22|±0.24|±0.17|
>
> **QQP Std (in %):**
> |Method|n=8|n=16|n=32|n=64|
> |-|-|-|-|-|
> |FT|±0.31|±0.52|±0.33|±0.44|
> |NTK|±0.11|±0.17|±0.15|±0.21|
> |SignGD|±0.21|±0.29|±0.23|±0.31|
> |OAK|±0.13|±0.19|±0.21|±0.23|
>
> Notably, all standard deviations are less than **0.0052 (±0.52%)**, ensuring the reported performance gaps are statistically significant.
>
> ---
> **[Q.4 Explanations on Technical Contributions of `Theorems 5.2, 5.4`]**
>
> Thanks for insightful question! The primary aim and innovation of `Theorems 5.2 and 5.4` are as follows:
>
> - **Aim of Theorems (`Section 5.1 & 5.2`):** These theorems provide formal guarantees for the accuracy of our OAK framework.
>
> - **Technical Innovations:** We highlight two key technical innovations in our proofs:
>     - Error Decomposition (`Appendix B.5, B.6`): Adaptive preconditioners are inherently cooedinate-wise and time-varying. To bnound this dynamics, we decompose the estimation error into Sampling Error (controlled by data) and Dynamic Error (controlled by training dynamics), which provides a **principled way to analyse gradient dynamics into standard kernel framework**.
>     - Bounding Cumulative Preconditioner (`Eq. 37, 43`): Adaptive optimizer’s state is cumulative over intermediate gradients, which are hard to estimate individually. To resolve this, we developed a recursive bounding technique using a specific identity ($1 = (1-\beta_1)\sum_{i=1}^{T}\beta_1^{T-i} + \beta_1^T$) which allows us to counteract intermediate gradient fluctuations, and **reduce intermediate gradients to a function of the total parameter drift $D_T$**.
>
> We have added discussion regarding these technical innovations in `Section 5`.
>
> ---
> **[Q.5 Explanation on "Avg. Acc" Meaning]**
>
> Yes, the "Avg. Acc." is the arithmetic mean of the accuracy across the respective row. Our intention is to provide a global metric over different tasks to demonstrate OAK’s superior performance. We have add an explicit definition of this metric for clarity.
>
> ---
>
> [1] lpNTK: Better Generalisation with Less Data via Sample Interaction During Learning, in ICLR
>
> [2] Dataset Meta-Learning from Kernel Ridge-Regression, in ICLR.
>
> [3] LENSLLM: Unveiling Fine-Tuning Dynamics for LLM Selection, in ICML.

---

> > ### Author Rebuttal · Reviewer_Tcew · 2026-04-04
> >
> > Thanks for the response.

---

### Official Review · Reviewer_RgN2 · 2026-03-21

**Soundness:** 2
**Presentation:** 3
**Significance:** 2
**Originality:** 3
**Overall Recommendation:** 5
**Confidence:** 3

**Summary:**

In this article, the authors adapt the computation of the neural tangent kernel to adaptive optimizers like Adam in fine-tuning tasks. For this, they propose the Optimize Aware Kernel (OAK) to modify the gradients on the test set by taking advantage of the first and second moment of the gradient on a training set. By taking the dot product between the gradients of data points in the training and test set respectively, they derive the OAK formulation. The authors provide theoretical upper bounds to the error between the true output of a model and a linearized version in Theorem 4.10. They also provide an upper bound on the deviation between the gradient at time $T$ and at time $0$, essentially capturing the difference with the kernel regime gradient. These two theorems show that their respective bounds become larger as $T \eta$ increases, where $\eta$ is the learning rate. The authors also introduce bounds on the change to the momentum and variance terms in preconditioning. These bounds are made tight by increasing the number of test set samples. An extensive set of fine-tuning experiments show that the OAK outperforms baselines on standard tasks from the GLUE benchmark.

**Compliance With Llm Reviewing Policy:**

Affirmed.

**Final Justification:**

Based on the rebuttal from the author, I have updated my score to 5: accept.

**Key Questions For Authors:**

- When introducing $\mathcal{L}(\xi; \theta)$, no dependence on $y$ is shown but it seems to depend on it. Can you add it or mention that it is implied for clarity?
- What is the convention used for computing the Jacobian? Are the number of columns the same as the dimension of $\theta$ or is it the number of rows? Check the following comment and also definition 4.1 for the inner product.
- Since $\nabla_{\theta} f(\xi; \theta)$ is a matrix, what is $\langle \nabla_{\theta} f(\xi_i; \theta), \nabla_{\theta} f(\xi_j; \theta) \rangle$? Is it simply taking the transpose of the matrix on the right and multiplying both matrices?
- In algorithm 1, why take the trace when computing $K_{ij}$? For muti-dimensional outputs to f, is the kernel $K_{NTK}(\xi_i, \xi_j)$ a matrix? Is the dimension output $o =1$? It seems to be the case given $\nabla^2_{\theta} f(\xi; \theta) \in \mathbb{R}^{d \times d}$.
- Lines 271-274, right column, as $T$ and $\eta$ increase, is it not rather the case that the upper bounds become looser? Unless these bounds are tight in some cases.
- For 2a) and 2b), the bounds increase to infinity, but does it imply that the errors truly become large? The results provide upper bounds but do you have theoretical reasons for believing your claim (e.g. tight bounds), or simply empirical evidence?
- Based on the previous point, how statistically significant are figures 2 and 3?
- In tables 4 and 5, are the values reported the accuracy on the task? Please add this mention to the caption. If it is another compatibility measure, where is it defined? Also, which architecture is used?

**Limitations:**

- Please see weaknesses above.
- The paper is lacking some discussion about the significance of results and comparison with the related literature.

**Strengths And Weaknesses:**

Strengths:
- Well written paper.
- Theoretical results are
- Extensive experimental results that show state-of-the-art performance fine-tuning tasks.
- No error detected in proofs.

Weaknesses:
- Some confusion regarding notation and other implicit details (see comments below).
- Theoretical results provide upper bounds on error terms, yet claim the increase in these bounds with $T \eta$ explains divergence from the NTK regime. Are these bounds theoretically tight? Figures 2 and 3 are unclear in terms of the values of $T$ and $\eta$, as well as the statistical significance of the discrepancy. I believe additional details are required to make the original claim stronger. For example, does the divergent trend continue for even larger values of $n$ across various tasks and architectures?

---

> ### Author Rebuttal · Authors · 2026-03-31
>
> Dear Reviewer RgN2:
>
> Thank you very much for your insightful feedback! We address each of your questions below.
>
> ---
> **[W.2 & Q.5 & Q.6 Tightness of `Theorem 4.10, 4.11`]**
>
> We thank the reviewer for the insightful question. We clarify that our bounds are **tight in order (Asymptotically tight)** w.r.t. $T\eta$. We analyze this in detail by decomposing the dependency chain of Error Terms in `Theorem 4.10, 4.11` as: $T\eta$ (Cumulative Steps) → $D_T$ (Parameter Drift) → $\\|E\\|,\\|R\\|$.
>
> - **1. Cumulative Steps vs. Parameter Drift** ($D_T = \mathcal{O}( \frac{T\eta}{1 - T\eta L_\theta})$)
> This directly comes from `Eq. 25` $D_T \le T\eta(\epsilon_g + L_\theta D_T)$, which is essentially applying the Lipschitz gradient `Assumption 4.3` to all training steps. `Eq. 25` further **degenerates to equality** when the optimization trajectory aligns with the maximum curvature $L_\theta$ of the loss-space. Therefore, the solution $\frac{x}{1-x}$ is tight.
> - **2. Parameter Drift vs. Error Terms** (e.g., $\\|E\\|_2 = \mathcal{O}(D_T) + \mathcal{O}(D_T^2)$)
> This dependency originates directly from fundamental calculus. For example, in feature drift $\\|E\\|_2$, the **$\mathcal{O}(D_T)$ term is dictated by the Mean Value Theorem**, which requires the change in gradients to scale at least linearly with $D_T$. Also, the **$\mathcal{O}(D_T^2)$ arises by directly applying standard Lipschitz Hessian (`Assumption 4.4`)**, which depicts the emerging curvature of loss space. Similarly, the linearization error $\\|R\\|_2$ follows the same derivation via the Taylor expansion remainder of the network $f$, and the orders of both cannot be further reduced.
>
> **Conclusion:** Combining the two dependencies, we have `Theorems 4.10 and 4.11` are **asymptotically tight** w.r.t. $T\eta$. We have added the tightness discussion in our paper.
>
> ---
> **[W.3 Divergent trend for larger n]**
> We clarify `Figure 2` uses $n$ as proxy for steps, as n scales, training steps T increase proportionally. We extended n to 2048 on MNLI (Qwen2.5-0.5B):
>
> |n|8|16|32|64|128|256|512|1024|2048|
> |-|-|-|-|-|-|-|-|-|-|
> |FT|33.5|34.0|34.4|35.5|46.4|53.5|58.6|62.5|65.2|
> |OAK|33.9|37.0|36.3|37.7|38.9|39.6|42.6|44.1|45.3|
> |Gap|-0.4|-3.0|-1.9|-2.2|7.5|13.9|16.0|18.4|19.9|
>
> Conclusion: Gap increases with even larger n, aligning with `Theorem 4.10 and 4.11`. We have updated `Figure 2` with these results.
>
> ---
> **[Q.1 Clarification on Notation $\mathcal{L}$]**
>
> Yes, the loss depends on the target $y$. We have updated the notation to $\mathcal{L}(ξ,y;θ)$ for mathematical clarity.
>
> ---
> **[W1 & Q.2 & Q.3 & Q.4 Clarification on Notation and Shapes]**
>
> We clarify the notation to resolve the concerns regarding dimensionality:
>
> - **Jacobian Convention:** We follow the **numerator layout**, where the Jacobian $J = \nabla_\theta f \in \mathbb{R}^{o \times d}$ ($o$ is the output dimension and $d$ is the parameter dimension). For the theoretical analysis in `Section 4`, we simplify the network output to a scalar ($o=1$, specifically the logit of the ground-truth class) to avoid 3rd-order tensors, which is a **standard simplification** in NTK literature [1][2].
> - **Frobenius Inner Product:** The inner product $\langle J_i, J_j \rangle$ in `Definition 4.1` refers to the **Frobenius inner product**, defined as $\text{Tr}(J_i J_j^T)$. This operation ensures that the kernel entry $K_{ij}$ is a scalar even for multi-dimensional ($o\times o$) outputs.
> - **Trace in Algorithm 1:** The trace operation in `Algorithm 1` implements the Frobenius inner product, collapsing the $o \times o$ matrix product into the scalar $K_{ij}$ required for kernel regression.
>
> We have clarified these notation details in the revised manuscript.
>
> ---
> **[Q.7 Clarification on Statistical Significance]**
>
> We thank the reviewer for the comment. All values reported in `Figure 2 and 3` are the average accuracy of 5 trials, we provide the detailed standard deviation (Std) for each point below:
>
> Table 1: Std for `Figure 2` (%)
> |Number of Sample $n$|8|16|32|64|128|256|512|
> |-|-|-|-|-|-|-|-|
> |Finetune|±.45|±.28|±.32|±.35|±.28|±.32|±.39|
> |OAK|±.15|±.09|±.18|±.15|±.19|±.21|±.16|
>
> Table 2: Std for `Figure 3` (%)
> |Shuffle Ratio|0|0.1|0.2|0.3|0.4|0.5|0.6|0.7|
> |-|-|-|-|-|-|-|-|-|
> |Finetune|±.42|±.45|±.48|±.31|±.53|±.50|±.52|±.55|
> |OAK|±.15|±.14|±.18|±.13|±.07|±.16|±.19|±.21|
>
> We have updated `Figure 2 and 3` with error bars in the revised version.
>
> ---
> **[Q.8 Clarification on Metrics]**
> Yes, the values in Table 4 and 5 are accuracies of RoBERTa on MNLI. We have updated the captions.
>
> ---
> [1] Wide Neural Networks of Any Depth Evolve as Linear Models Under Gradient Descent, in NeurIPS.
>
> [2] A Kernel-Based View of Language Model Fine-Tuning, In ICML.

---

> > ### Author Rebuttal · Reviewer_RgN2 · 2026-04-02
> >
> > My comments have been properly addressed. For this reason, I will adjust my score to 5: accept.

---

> > > ### Author Response · Authors · 2026-04-04
> > >
> > > Dear Reviewer RgN2,
> > >
> > > Thank you very much for your positive acknowledgement. We are glad that our rebuttal has fully addressed your concerns.
> > >
> > > We also appreciate your note that you would kindly adjust your score to 5 (Accept). If convenient, we would be grateful if you could reflect this score update in the OpenReview system as well. We sincerely thank you again for your constructive feedback!
> > >
> > >
> > > Best regards,
> > >
> > > Authors of Submission 6811

---

### Decision · Program_Chairs · 2026-04-30

**Decision:**

Accept (regular)

**Comment:**

This paper proposes the Optimizer Aware Kernel (OAK), which extends the empirical Neural Tangent Kernel framework by incorporating a data-driven static estimate of the adaptive optimizer's preconditioner into the kernel construction. The paper derives explicit error bounds on the linearization error and feature drift during fine-tuning (Theorems 4.10 and 4.11) and on the preconditioner estimation itself (Theorems 5.2 and 5.4), and reports experiments on RoBERTa and Qwen2.5 across GLUE tasks.

**Strengths.** Reviewers agree that extending NTK-based analysis to adaptive optimizers is an important and timely direction (Tcew, XwpW). The two theoretical components are internally consistent — the preconditioner estimation error and the kernel-regime collapse bounds are driven by the same factors, which is uncommon for this area (XwpW). The static-preconditioner estimation is a reasonable way to sidestep dependence on historical gradients (XwpW). Presentation is clear with structured outlines and explicit takeaways per theorem (3aHA, Tcew), and OAK outperforms both standard eNTK and the SignGD-NTK baseline (Malladi et al., 2023) in low-data regimes across encoder-based and decoder-only architectures.

**Weaknesses.** Most reviewer concerns were resolved during rebuttal. Three items remain appropriate for camera-ready attention. First, Theorems 4.10-4.11 are proved under GD while experiments use Adam; the rebuttal extends the bounds to bounded preconditioners with eigenvalues in $[c_1, c_2]$, showing the drift and error bounds scale by constants — this should be incorporated in the main text with an explicit discussion of the stricter condition $c_2 T \eta L_\theta < 1$ that replaces the GD condition (XwpW). Second, the training Gram-matrix construction and kernel-regression formula clarified in rebuttal should enter Section 3 (3aHA). Third, Section 4's theoretical analysis adopts scalar-output ($o=1$) convention while Algorithm 1 uses the Frobenius inner product for multi-output Jacobians; a brief note reconciling the two should be added (RgN2).

**Decision.** I recommend **accept**. All four reviewers support acceptance with post-rebuttal distribution $5, 4, 4, 4$, all fully_resolved. The authors should incorporate the rebuttal material above into the camera-ready, along with the controlled-$T$ ablation, the wall-clock and memory measurements, and the LAMA generation experiment from the rebuttal, and the kernel-based data-selection experiment demonstrating practical utility.